# Pairwise Optimal Transports for Training All-to-All Flow-Based Condition Transfer Model

**Kotaro Ikeda**
The University of Tokyo
Preferred Networks, Inc.*
kotaro-ikeda@g.ecc.u-tokyo.ac.jp

**Masanori Koyama**
The University of Tokyo
Preferred Networks, Inc.

**Jinzhe Zhang**
Preferred Networks, Inc.

**Kohei Hayashi**
The University of Tokyo
Preferred Networks, Inc.

**Kenji Fukumizu**
The Institute of Statistical Mathematics
Preferred Networks, Inc.

## Abstract

In this paper, we propose a flow-based method for learning all-to-all transfer maps among conditional distributions that approximates pairwise optimal transport. The proposed method addresses the challenge of handling the case of continuous conditions, which often involve a large set of conditions with sparse empirical observations per condition. We introduce a novel cost function that enables simultaneous learning of optimal transports for all pairs of conditional distributions. Our method is supported by a theoretical guarantee that, in the limit, it converges to the pairwise optimal transports among infinite pairs of conditional distributions. The learned transport maps are subsequently used to couple data points in conditional flow matching. We demonstrate the effectiveness of this method on synthetic and benchmark datasets, as well as on chemical datasets in which continuous physical properties are defined as conditions. The code for this project can be found at https://github.com/kotatumuri-room/A2A-FM

## 1 Introduction

Recent advances in generative modeling have been largely driven by the theory of dynamical systems, which describes the transport of one probability measure to another. Methods such as diffusion models [40, 18] and flow matching (FM, [31, 32, 2]) achieve this transport by leveraging stochastic or ordinary differential equations to map an uninformative source distribution to target distribution(s). Incorporating conditional distributions is a critical aspect of these dynamical generative models, as this enables the generation of outputs with specific desired attributes. Considerable research has been dedicated to designing mechanisms to condition these models. For example, in image and video generation, text prompts are commonly used to guide the output toward specified content [12, 17]. Similarly, in molecular design, conditioning on physical properties enables the generation of molecules with target characteristics [46, 25].

This paper focuses on *condition transfer* (Fig. 1 (a)), an essential task in conditional generative modeling whose goal is to transport an arbitrary conditional distribution to another. In applications, it is often used to modify the specific attributes or conditions of a given instance while preserving its other features. Applications span various domains; for example, in computer vision, image style transfer has been an active area of research [16, 48], whereas chemistry, modifying molecules to achieve desired physical properties is vital for exploring new materials and drugs [23, 25].

---

*This work was partially done while Kotaro was an intern in Preferred Networks.

39th Conference on Neural Information Processing Systems (NeurIPS 2025).

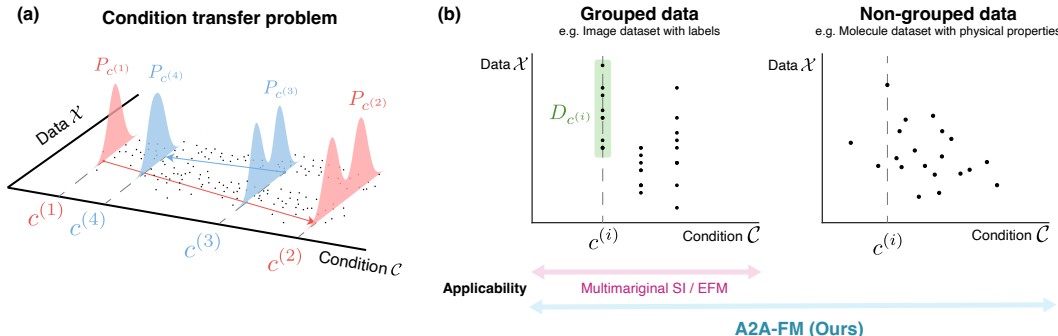

Figure 1: (a) The task is to transport $x_{\mathrm{src}} \sim P_{c_{\mathrm{src}}}$ to generate $x_{\mathrm{targ}} \sim P_{c_{\mathrm{targ}}}$ for arbitrary $(c_{\mathrm{src}}, c_{\mathrm{targ}})$ pair, where $P_c$ denotes the conditional distribution. Red and blue arrows respectively represent the case of $(c_{\mathrm{src}}, c_{\mathrm{targ}}) = (c^{(1)}, c^{(2)})$ and $(c_{\mathrm{src}}, c_{\mathrm{targ}}) = (c^{(3)}, c^{(4)})$. (b) Left: Grouped data is the type of dataset that can be grouped into subsets $D_{c^{(i)}}$ of large size, whose members are i.i.d. samples from $P_{c^{(i)}}$. Many condition transfer methods including Multimariginal Stochastic Interpolants (SI) [3] and Extended Flow Matching (EFM) [21] leverage this data format. Right: In non-grouped data, a sample corresponding to a given condition can be unique. Proposed method, A2A-FM, can learn condition transfer on both cases in the form of pairwise optimal transport. (see Section 4 for comparision with related works).

Flow-based generative models have been explored for condition transfer. Among others, Albergo et al. [2] proposed Stochastic Interpolants (SI), a flow-based method that interpolates between two distributions. Tong et al. [41] introduced OT-CFM, which uses optimal transport [43] to couple the data points in minibatches for conditional FM, and applied it to tasks such as image style transfer and single-cell expression data. Meanwhile, recent methods, such as Multimarginal SI [3] and Extended Flow Matching (EFM, [21]) have extended FM approaches to support all-to-all(multiway) transports between multiple conditional distributions.

A challenge in condition transfer arises particularly when dealing with *continuous* condition variables. This scenario encompasses critical scientific applications, such as modifying molecules based on continuous physical properties. Such continuous conditions can pose the difficulty that each condition $c$ may be associated with only a single observation $x$, resulting in limited information for each conditional distribution. Moreover, the situation may involve an infinite number of source-target condition pairs; applying a method designed for two distributions to all the pairs would be computationally infeasible. Most existing approaches cannot resolve these difficulties because they assume what we call *grouped data*, a type of dataset in which each observed condition is associated with a sufficiently large number of samples (Fig. 1 (b)). To the best of our knowledge, no method that can learn the *all-to-all* transfer model for condition transfer scalably on general datasets of observation-condition pairs, including non-grouped ones.

This paper proposes All-to-All Flow-based Transfer Model (A2A-FM), a novel method that solves condition transfer tasks on general datasets (even in non-grouped data setting) by simultaneously learning a *family of flows* that approximates the optimal transport (OT) between any pair of conditional distributions. Inspired by the technique of [9, 27], we develop a novel cost function with a theoretical guarantee that empirical couplings converge to the pairwise OT in the infinite sample limit.

The main contributions of this paper are as follows.

- A2A-FM, an FM-based method, is proposed to learn pairwise optimal transport maps across all conditional distributions from general datasets, including both grouped data and non-grouped data, irrespective of whether the conditional variables are continuous.
- We introduce a novel cost function for coupling samples and prove that the coupling achieves the pairwise optimal transports in the infinite sample limit.
- A2A-FM is applied to a chemical problem of altering the target attributes of a molecule while preserving the other structure, and it demonstrates competitive and efficient performance.

## 2 Preliminaries

Suppose that we have a joint distribution $P$ on the space $\mathcal{X} \times \mathcal{C}$, where $\mathcal{X} \subset \mathbb{R}^{d_x}$ and $\mathcal{C} \subset \mathbb{R}^{d_c}$ denote the data space and the condition space, respectively. Let $\{P_c | c \in \mathcal{C}\}$ denote the family of conditional distributions parameterized by $c$. Our method belongs to the family of Flow Matching (FM) frameworks [31, 32, 2]. Therefore, we first review FM and conditional optimal transport in the context of conditional generation, before turning to our condition-transfer task. Throughout the paper, we use the notation $x \sim Q$ to denote that $x$ follows the distribution $Q$, and $\dot{x}(t)$ to denote the time derivative of a smooth curve $x(t)$. The set of integers $\{1, \dots, N\}$ is denoted by $[1 : N]$.

### 2.1 Flow Matching (FM)

FM was originally developed to learn a transport from a fixed, possibly uninformative source distribution, such as the normal distribution, to a target distribution, for which samples $\{x^{(j)}\}$ are available for training. Below, we review a standard FM method, but adapted to learn the conditional distributions. The source distribution $P_\emptyset$ is transported to the target $P_c$ via an ODE

$$\dot{x}_c(t) = v(x_c(t), t|c), \quad x_c(0) \sim P_\emptyset, \tag{1}$$

so that $\Phi_{c,1}(x(0))$ follows $P_c$, where $\Phi_{c,t}(x(0)) := x_c(t)$ is the flow defined by the ODE. FM learns the vector field $v(x_c(t), t \mid c)$ via neural networks with the parameter set $\theta$, using the loss function

$$L(\theta) = \mathbb{E}_{\psi_c, c, t}[\|v_\theta(\psi_c(t), t|c) - \dot{\psi}_c(t)\|^2], \tag{2}$$

where $\psi_c : [0, 1] \to \mathcal{X}$ is a random path such that $\psi_c(0)$ and $\psi_c(1)$ follow $P_\emptyset$ and $P_c$, respectively.

### 2.2 Optimal Transport (OT) and OT-based FM methods

A popular method for constructing the random paths $\psi_c(t)$ is minibatch optimal transport (OT-CFM, [41, 36]), which aims to approximate the ODE of optimal transport.

Specifically, OT-CFM uses a linear path $\psi_c(t) = t\psi_c(1) + (1 - t)\psi_c(0)$. The key to this approach lies in how the start point $\psi_c(0) \sim P_\emptyset$ and the end point $\psi_c(1) \sim P_c$ are coupled. OT-CFM utilizes a coupling based on the optimal transport map between these two distributions. The theoretical basis for this is the optimal transport problem. Given two distributions $Q_1, Q_2$ on $\mathcal{X}$, the Kantorovich formulation of the OT problem is as follows:

$$\inf_{\Pi \in \Gamma(Q_1, Q_2)} \int_{\mathcal{X}^2} \|x_1 - x_2\|^2 d\Pi(x_1, x_2), \tag{3}$$

where $\Gamma(Q_1, Q_2)$ is the set of all joint distributions of $(x_1, x_2)$ whose marginals are $Q_1$ and $Q_2$, respectively. The square root of the infimum is known as the 2-Wasserstein distance, $W_2(Q_1, Q_2)$ [44].

When $Q_1$ satisfies certain conditions (e.g., is absolutely continuous), this problem can be reduced to the Monge formulation, which seeks a transport map $T : \mathcal{X} \to \mathcal{X}$ [44]:

$$\inf_{T:\mathcal{X} \to \mathcal{X}} \int_{\mathcal{X}} \|x_1 - T(x_1)\|^2 dQ_1(x_1), \tag{4}$$

where the infimum is taken over all maps $T$ that satisfy the push-forward condition $Q_2 = T\#Q_1$.

Denoting the optimal transport map that solves Eq. (4) for $Q_1 = P_\emptyset$ and $Q_2 = P_c$ as $T_{\emptyset \to c}$, the ideal random path for OT-CFM is defined by the deterministic mapping $\psi_c(1) = T_{\emptyset \to c}(\psi_c(0))$, which gives

$$\psi_c(t) = t\, T_{\emptyset \to c}(\psi_c(0)) + (1 - t)\, \psi_c(0). \tag{5}$$

In practice, however, this map $T_{\emptyset \to c}$ is unknown. Therefore, OT-CFM approximates this optimal coupling at the minibatch level. The sampling process consists of two steps: (i) drawing independent batches $(\{x_\emptyset^{(i)}\}, \{x_c^{(j)}\})$ from $P_\emptyset$ and $P_c$, respectively, and (ii) using a finite-sample OT algorithm, such as [10], to obtain a coupled pair $(x_\emptyset, x_c)$ between these batches. This pair $(\psi_c(0), \psi_c(1)) = (x_\emptyset, x_c)$ is then used to construct the linear path $\psi_c(t)$.

## 2.3 Conditional Optimal Transport (COT) for conditional generation

Some recent work [9, 28, 4, 20] proposed an algorithm and theory of conditional optimal transport(COT) that minimizes *conditional Wasserstein distance*. Using a dataset $\{(x^{(j)}, c^{(j)})\}$, their methods learn a transport from the conditional source $P_{\emptyset,c}$ to the conditional target distributions $P_c$ on the joint space $\mathcal{X} \times \mathcal{C}$. When using empirical OT in such a situation, a key challenge is that the condition-wise optimal coupling from conditional source to conditional target does not necessarily converge to the minimizer of conditional Wasserstein distance (see Example 9, [9]). To solve this problem, several works proposed a coupling algorithm on $\mathcal{X} \times \mathcal{C}$ that provably converges to the optimal transport for each $c$ [7, 20, 9]. Their objective is to minimize the following cost, where $\pi$ ranges over $\mathfrak{S}_N$, the symmetric group of $[1 : N]$.

$$\sum_{i=1}^{N} \|x_1^{(i)} - x_\emptyset^{\pi(i)}\|^2 + \beta \|c_1^{(i)} - c_\emptyset^{\pi(i)}\|^2. \tag{6}$$

Here, the two batches $B_\emptyset = \{(x_\emptyset^{(i)}, c_\emptyset^{(i)})\}_{i=1}^{N}$ and $B_1 = \{(x_1^{(j)}, c_1^{(j)})\}_{j=1}^{N}$ are drawn from the source-condition joint distribution $P_\emptyset$ and data distribution $P$, respectively. Chemseddine et al. [9] have shown that, if we regard the optimal coupling $\pi_\beta^*$ as the joint distribution $\Pi_\beta^* := \sum_i \delta_{(x_1^{(i)}, c_1^{(i)}) \times (x_\emptyset^{\pi_\beta^*(i)}, c_\emptyset^{\pi_\beta^*(i)})}$ on $(\mathcal{X} \times \mathcal{C}) \times (\mathcal{X} \times \mathcal{C})$, then with increasing $\beta$ and sample size $N$ (up to subsequence), the distribution $\Pi_\beta^*$ converges to a joint distribution supported on $\{(x, c, x', c) \in (\mathcal{X} \times \mathcal{C}) \times (\mathcal{X} \times \mathcal{C})\}$ that achieves the *conditional 2-Wasserstein distance*

$$\mathbb{E}_c[W_2^2(P_{\emptyset,c}, P_c)], \tag{7}$$

where $P_{\emptyset,c}$ and $P_c$ denote the conditional distributions given $c$ for the source $P_\emptyset$ and target $P$, respectively. Because (7) is the average 2-Wasserstein distance of the conditional distributions given $c$, the above result implies that the algorithm with the cost (6) constructs a desired coupling that realizes the optimal transport per condition for a large sample limit.

# 3 Purpose and Method

In this section, we first restate the goal of A2A-FM and describe its applicability to general dataset. We then introduce the training objective and provide theoretical support for the algorithm.

**Pairwise optimal transports**: The task of condition transfer for the family of conditional distributions $\{P_c | c \in \mathcal{C}\}$ can be cast as the problem of learning the $(c_1, c_2)$-parameterized transports $T_{c_1 \to c_2}$ from $P_{c_1}$ to $P_{c_2}$. A2A-FM is a method to learn *pairwise optimal transports* $\{T_{c_1 \to c_2} \mid c_1, c_2 \in \mathcal{C}\}$ that minimize the transport cost. More formally, writing the transport $T_{c_1 \to c_2}$ by its induced joint distribution $P_T(\cdot | c_1, c_2)$ on $\mathcal{X} \times \mathcal{X}$ with marginals $P_{c_1}$ and $P_{c_2}$(Kantorovich formulation), A2A-FM's goal is to learn the maps that simultaneously minimize the *pairwise transport cost*

$$\int_{\mathcal{X}^2} \|x_1 - x_2\|^2 dP_T(x_1, x_2 | c_1, c_2) \quad \text{for all } (c_1, c_2). \tag{8}$$

Pairwise transport cost (8) is the transport cost between the conditional distributions $P_{c_1}$ and $P_{c_2}$.

**Applicability to general dataset**: Most importantly, the scope of A2A-FM covers any type of dataset $D = \{(x^{(i)}, c^{(i)})\}$ drawn i.i.d from the joint distribution $P$, regardless of whether the data is grouped or non-grouped. While some recent methods [21, 2, 6] also aim to learn all-to-all condition transports, they require the more restrictive *grouped data* to obtain the information about $P_c$ and to compute its coupling to other distributions in the dataset. The significance of A2A-FM lies in the capability of learning the pairwise OT also from *non-grouped* data (Fig. 1 (b)), which arises naturally with continuously valued conditions in many scientific applications.

## 3.1 Training objectives and Procedure

Aligned with the OT-CFM [41], we propose a flow-based method through minibatch OT to achieve pairwise optimal transport. To learn $T_{c_1 \to c_2}$, we use $(c_1, c_2)$-parametrized ODE like (1):

$$\dot{x}_{c_1,c_2}(t) = v(x_{c_1,c_2}(t), t | c_1, c_2), \quad \text{where } x_{c_1,c_2}(0) \sim P_{c_1}, \ x_{c_1,c_2}(1) \sim P_{c_2}. \tag{9}$$

---

**Algorithm 1** Training of A2A-FM

---

**Input:** (i) Dataset of sample-condition pairs $D := \{(x^{(i)}, c^{(i)})\}$, where each $x^{(i)} \in \mathcal{X}$ is sampled from $P_{c^{(i)}}$. (ii) A parametric model of a vector field $v_\theta : \mathcal{X} \times [0, 1] \times \mathcal{C} \times \mathcal{C} \to \mathcal{X}$. (iii) The scalar parameter $\beta$. (iv) An algorithm OPTC for optimal coupling.
**Return:** The parameter $\theta$ of $v_\theta$

1: **for** each iteration **do**
    # Step 1: Compute the coupling
2:    Subsample batches $B_1 = \{(x_1^{(i)}, c_1^{(i)})\}_{i=1}^N$, $B_2 = \{(x_2^{(i)}, c_2^{(i)})\}_{i=1}^N$ from $D$.
3:    Minimize (11) about $\pi_\beta^*$ over $N$ indices by OPTC
    # Step 2: Update $v_\theta$
4:    Sample $t_i \sim \mathbf{unif}[0, 1]$, $i \in [1 : N]$.
5:    Update $\theta$ by $\nabla_\theta L(\theta)$ with
    $L(\theta) = \sum_{i=1}^N \|v_\theta(\psi_i(t_i), t_i | c_1^{(i)}, c_2^{\pi_\beta^*(i)}) - \dot{\psi}_i(t_i)\|^2$ where $\psi_i(t) = (1-t)x_1^{(i)} + tx_2^{\pi_\beta^*(i)}$.
6: **end for**

---

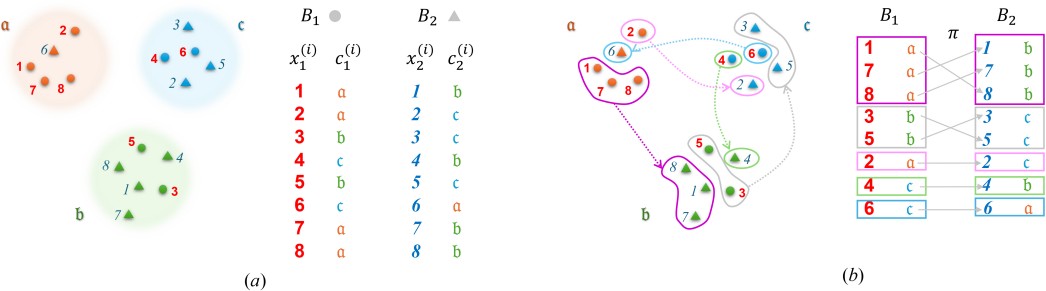

Figure 2: (a) Batches $B_1, B_2$ drawn independently from $P$ on $\mathcal{X} \times \mathcal{C}$, where $\mathcal{C} = \{a, b, c\}$. (b) Couplings between $B_1$ and $B_2$ by (11). With large $\beta$, the cost favors $\pi$ such that $(c_1^{(i)}, c_2^{(i)}) = (c_1^{\pi(i)}, c_2^{\pi(i)})$.

The model for the vector field $v : \mathcal{X} \times [0, 1] \times \mathcal{C} \times \mathcal{C} \to \mathbb{R}^{d_x}$ is trained with random path $\psi_{c_1, c_2}$, whose endpoints are drawn from the coupled points in two minibatches, as explained below. The training objective for $v$ is given by

$$\mathbb{E}_{\psi_{c_1, c_2}, c_1, c_2, t}\left[\|v(\psi_{c_1, c_2}(t), t | c_1, c_2) - \dot{\psi}_{c_1, c_2}(t)\|^2\right]. \tag{10}$$

The crux of A2A-FM is how we construct the coupling that defines $\psi_{c_1, c_2}$. We begin by sampling two independently chosen batches $B_1 = \{(x_1^{(i)}, c_1^{(i)})\}_{i=1}^N$, $B_2 = \{(x_2^{(i)}, c_2^{(i)})\}_{i=1}^N$ of the same size $N$ from the dataset $D$. The objective function of the coupling is

$$\sum_{i=1}^N \|x_1^{(i)} - x_2^{\pi(i)}\|^2 + \beta\left(\|c_1^{(i)} - c_1^{\pi(i)}\|^2 + \|c_2^{(i)} - c_2^{\pi(i)}\|^2\right), \tag{11}$$

where $\pi$ runs over $\mathfrak{S}_N$. Letting $\pi_\beta^*$ be the minimizer of (11), we define the path $\psi_i = \psi_{c_1, c_2}$ with $(c_1, c_2) = (c_1^{(i)}, c_2^{\pi_\beta^*(i)})$ for $i \in [1 : N]$ by

$$\psi_i(t) = (1-t)x_1^{(i)} + tx_2^{\pi_\beta^*(i)}.$$

To transport a sample $x_{c_1} \sim P_{c_1}$ to a sample in $P_{c_2}$ using the trained $v$, we follow the same procedure as in standard FM and solve the ODE (9) forward from $t = 0$ to $t = 1$ with initial condition $x_{c_1, c_2}(0) = x_{c_1}$. The algorithmic training procedure is summarized in Algorithm 1.

### 3.2 Intuition behind the objective

The objective function (11) controls the continuity of $P_c$ with respect to $c$ in the transfer maps through a balance of the hyperparameter $\beta$ and the sample size $|D| = N$. When there is only one sample $x_c$

per each $c$, it is impossible to learn $P_c$ separately, let alone the optimal transport between $P_{c_1}$ and $P_{c_2}$. However, if $\beta$ is small enough, data points $x_{c_1} \sim P_{c_1}$ and $x_{c_2} \sim P_{c_2}$ can be coupled for almost any $c_1$ and $c_2$, so that the information between $P_{c_1}$ and $P_{c_2}$ is shared. Conversely, as $\beta \to \infty$, (11) would force $c_k^{(i)}$ to be close to $c_k^{\pi(i)}$, requiring that for each pair of $(c_1, c_2)$, the sample $x_1^{(j)}$ with condition $c_1^{(j)}$ near $c_1$ is coupled with $x_2^{(j)}$ with condition $c_2^{(j)}$ near $c_2$. This would allow the samples with similar $c$ values to be transported collectively.

We illustrate our way of coupling when $\mathcal{C}$ is finite (Fig. 2), where the functionality of (11) is more intuitive. In that case, for $\beta \to \infty$, the second term in (11) would diverge to infinity unless it is zero. However, for $|\mathcal{C}| < \infty$, as the sample size $N$ increases, there will be many nontrivial permutations $\pi$ with $c_k^{(i)} = c_k^{\pi(i)}$ ($k = 1, 2$), which makes the second term zero. Therefore, the minimum of (11) is attained by a permutation in $\mathfrak{S}^o := \{\pi \in \mathfrak{S}_N \mid c_k^{(i)} = c_k^{\pi(i)} \text{ for all } i \in [1 : N], k = 1, 2\}$. We can see the function of $\mathfrak{S}^o$ more clearly by partitioning $[1 : N]$ into the subsets closed within $\mathcal{C}$; $[1 : N] = \biguplus_{(c_1, c_2) \in \mathcal{C}^2} U_{(c_1, c_2)}$, where $U_{(c_1, c_2)} := \{i \in [1 : N] \mid c_k^{(i)} = c_k, (k = 1, 2)\}$. For example, in Fig. 2, the subset $\mathfrak{S}^o$ consists of permutations that act separately on $\{1, 7, 8\}$, $\{3, 5\}$, $\{2\}$, $\{4\}$, and $\{6\}$. Consequently, a permutation in $\mathfrak{S}^o$ is decomposed into permutations within $U_{(c_1, c_2)}$, and $\mathfrak{S}^o$ is the the product of symmetric groups of $U_{(c_1, c_2)}$. The minimization (11) is reduced to the sum of

$$\sum_{i \in U_{(c_1, c_2)}} \|x_1^{(i)} - x_2^{\pi(i)}\|^2,$$

where each summand is the standard cost of transporting data from class $c_1$ in $B_1$ to class $c_2$ in $B_2$. Therefore, the optimal $\pi_\beta^*$ achieves the optimal coupling among $\{x_1^{(i)} \mid i \in U_{(c_1, c_2)}\} \to \{x_2^{\pi(i)} \mid i \in U_{(c_1, c_2)}\}$. In Fig. 2, we see $U_{(\mathfrak{a}, \mathfrak{b})} = \{1, 7, 8\}$, and the optimal $\pi_\beta^*$ would optimally transport $\{x_1^{(1)}, x_1^{(7)}, x_1^{(8)}\}$ in Class $\mathfrak{a}$ to $\{x_2^{(1)}, x_2^{(7)}, x_2^{(8)}\}$ in Class $\mathfrak{b}$. As $N$ increases, each $U_{(c_1, c_2)}$ contains a larger number of indices, so that $\pi_\beta^*$ better approximates the OT from $P_{c_1}$ to $P_{c_2}$ for every $(c_1, c_2)$. Interestingly, although we have discussed the case of finite $\mathcal{C}$ in this section, this argument extends to the general case of continuous $\mathcal{C}$. We elaborate on this fact in the next subsection.

### 3.3 Theoretical guarantee

We show that the coupling given by (11) converges to pairwise OT. We state the result informally here and defer the formal result to Appendix A.

**Proposition 3.1** (Informal). *Let $\Pi_\beta^*$ be the joint distribution on $(\mathcal{X} \times \mathcal{C}) \times (\mathcal{X} \times \mathcal{C})$ defined by the coupling $\pi_\beta^*$ that minimizes (11), that is*

$$\Pi_\beta^* = \sum_{i=1}^N \delta_{((x_1^{(i)}, c_1^{(i)}), (x_2^{\pi_\beta^*(i)}, c_2^{\pi_\beta^*(i)}))}.$$

*Then, for any sequence $\beta_k \to \infty$, there exists an increasing sequence of the sample size $N_k$ such that $\Pi_{\beta_k}^*$ converges to $\Pi^*$ for which $\Pi^*(\cdot, \cdot \mid c_1, c_2)$, the corresponding conditional distribution of $(x_1, x_2)$ given $(c_1, c_2)$, is a joint distribution on $\mathcal{X} \times \mathcal{X}$ that achieves below for almost every $(c_1, c_2)$ :*

$$\int_{\mathcal{X}^2} \|x_1 - x_2\|^2 d\Pi^*(x_1, x_2 | c_1, c_2) = W_2^2(P_{c_1}, P_{c_2}). \tag{12}$$

We will show experimentally in Section 5 that, with the objective function (11), A2A-FM learns an approximate pairwise optimal transport even on the generic dataset drawn from joint distribution $P$.

## 4 Related Works

In addition to the direct application of optimal transport to transfer tasks [11, 35], other works such as [41, 32] used flow models to transfer between two distributions with different attributes (e.g., images of smiling faces vs. non-smiling). Kapuśniak et al. [24] also incorporated the geometry of the data manifold to the flow. Although one can apply these methods to many pairs of conditions individually, such a strategy is computationally inefficient for the all-to-all condition transfer task.

In this regard, Albergo et al. [3] presented Multimarginal SI, which uses the principle of generalized geodesic [5]. In this framework, one produces $T_{c_1 \to c_2}$ for arbitrary $c_1, c_2$ as a linear ensemble of

| (a) Grouped data (Fig. 3 (a)) | | (b) Non-grouped data (Fig. 3 (b)) | |
| --- | --- | --- | --- |
| Method | MSE from pairwise OT | Method | MSE from pairwise OT |
| Ours ($\pi_\beta^*$) | $\mathbf{(5.81 \pm 2.22) \times 10^{-2}}$ | A2A-FM (Ours) | $\mathbf{(1.51 \pm 0.17) \times 10^{-2}}$ |
| Generalized geodesic | $(1.03 \pm 0.04) \times 10^{0}$ | Partial diffusion | $(6.77 \pm 0.14) \times 10^{-2}$ |
| | | Multimarginal SI | $(4.90 \pm 0.28) \times 10^{-2}$ |

Table 1: The discrepancy from pairwise OT for synthetic data. The error in the table shows the standard deviation over 10 evaluation runs.

the optimal transports $T_{\emptyset \to c^{(k)}}$ from an arbitrary barycentric source $P_\emptyset$ to $P_{c^{(k)}}$. By design, this approach requires grouped data for the learning of $T_{\emptyset \to c^{(k)}}$. Isobe et al. [21] take another approach using a matrix field for the system of continuity equations, from which an arbitrary $T_{c_1 \to c_2}$ can be derived. For the learning of the matrix field, grouped data is again required. In another related note, [6] recently proposed a method that can transport *any* source distribution to a target distribution by encoding the source population itself, and conditioning the vector field with the encoded population. This approach again requires grouped data for the learning of population embedding.

Meanwhile, Manupriya et al. [33] uses COT [9, 28] to learn the OT map from $P_c$ to $Q_c$ for each $c$ when given the jointly sampled dataset of $(x, c) \sim P$ and $(y, c) \sim Q$. They handle non-grouped data, but they cannot create arbitrary $T_{c_1 \to c_2}$. Although the proof technique in COT is an inspiration for our proof, COT pursues *conditional generation* as opposed to the fundamentally different task of all-to-all *condition transfer*. In this regard, we emphasize that the objective (11) is fundamentally different from the one used in conditional OT (6). Note that, directly applying the method of similar nature as COT by a simple replacement of a single instance $c$ with $(c_1, c_2)$ would fail to learn a map between $c_1 \neq c_2$; if we learn COT for $v(\cdot|c_1, c_2)$ via (6) with $(c_1, c_2)$ in place of $(c_\emptyset, c_1)$, the model will only learn transfers between a pair of $c_1$ and $c_2$ that are close to each other.

## 5 Experiments

In this section, we present a series of experiments to validate our theoretical claims in Section 3, as well as the effectiveness of our model on a real-world dataset. The methods we use for comparison include (1) partial diffusion [25], a method that adds limited noise to a source sample and then denoises using classifer-free guidance to obtain a target-conditioned sample while preserving smilarity to the input (2) Multimarginal SI (see Section 4), and (3) an application of OT-CFM in which we use the sample dataset itself as the source $P_\emptyset$. In non-grouped settings, we generally choose $\beta = N^{1/(2d_c)}$, where $d_c$ is the dimension of $\mathcal{C}$ (see Appendix A.3 for theoritical backgrounds for this rate).

### 5.1 Synthetic Data (grouped and non-grouped data)

We demonstrate with synthetic data that both the coupling $\pi_\beta^*$ used to make the supervisory paths and the trained vector fields $v$ in A2A-FM approximate the pairwise optimal transport.

**Grouped data:** We compared the supervisory vector fields $\dot{\psi}_{c_1, c_2}$ of both A2A-FM and generalized geodesics against the pairwise OT on a synthetic dataset of three conditional distributions, each having the distribution of two-component Gaussian mixtures (Fig. 3 (a)). Note that A2A-FM more closely resembles the numerical approximation of the true pairwise OT [13]. Denoting by $(x_1, T_{c_1 \to c_2}(x_1)) = (\dot{\psi}_{c_1, c_2}(0), \dot{\psi}_{c_1, c_2}(1))$ the coupling of $P_{c_1}, P_{c_2}$ by a given method, we also validated this result quantitatively with $\mathbb{E}_{x_1, c_1, c_2}[\|T_{c_1 \to c_2}(x_1) - T_{c_1 \to c_2}^{\mathrm{OT}}(x_1)\|^2]$ (Table 1 (a)), where $T_{c_1 \to c_2}^{\mathrm{OT}}$ is the ground-truth pairwise optimal transport map that minimizes (8).

**Non-grouped data:** We compared the transfer map of A2A-FM against ground-truth pairwise OT, partial diffusion, and Multimarginal SI on non-grouped data. The dataset consisted of samples from a 2D polar coordinate quadrant ($r \in [1, 2]$, $\theta \in [0, \pi/2]$), where $\theta$ was a continuous condition and $P_\theta$ was uniform along $r$. This represents a non-grouped data. For Multimarginal SI, we discretized $\theta$ into $K = 5$ bins because it can only handle grouped data. See also Appendix B. We visualize the learnt transport in Fig. 3 (b). Partial diffusion produced nearly random couplings. Multimarginal SI failed to generate the target distribution's marginals because of discretization. Quantitative evaluations in Table 1 (b) also confirm that A2A-FM approximates pairwise OT more accurately than the rivals.

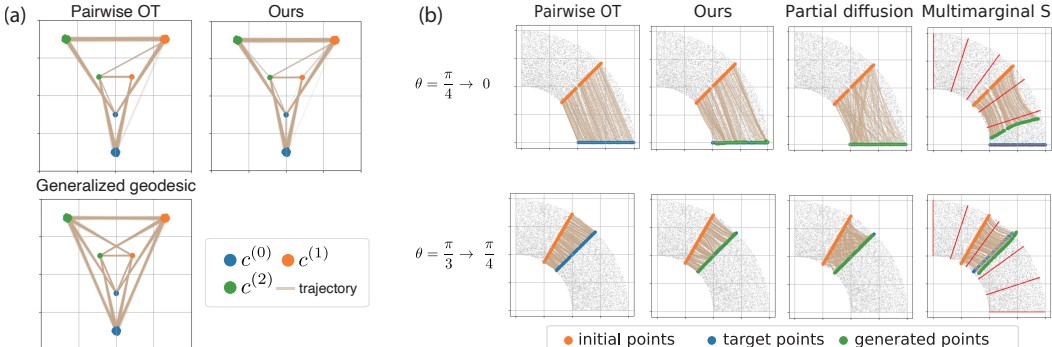

Figure 3: (a) Results for grouped data. The sample size was $10^3$ in each of 3 conditions $\{c^{(0)}, c^{(1)}, c^{(2)}\}$ and $\beta = 10^4$. (b) Results for non-grouped data. The gray points in the background show samples from the training dataset. The presented pairwise OT is a numerical approximation by [13]. The red lines in the right column shows the bins for training Multimarginal SI.

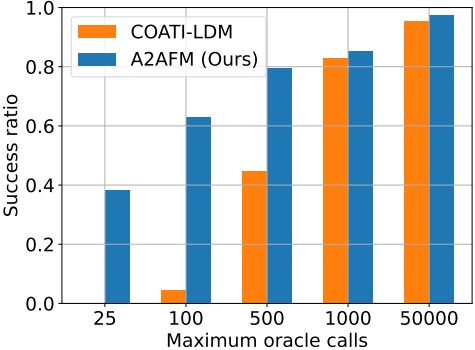

Figure 4: Sampling Efficiency of A2A-FM and the partial diffusion model of [26].

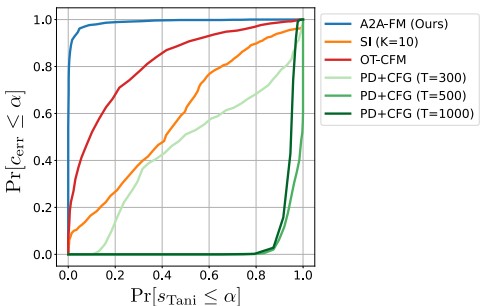

Figure 5: Sampling Efficiency Curve for LogP-TPSA benchmark. See Appendix B.3 and Table 3 for notation. $K$ is the number of discretization bins.

## 5.2 Nearby Sampling for Molecular Optimization

Molecule design is a multi-constraint task: a *lead* must bind strongly and selectively while remaining drug-like and nontoxic. Random candidates rarely meet every criterion, so chemists traditionally resort to scaffold hopping—generating many close analogs of existing molecules in the hope that some retain favorable properties while mitigating undesirable ones. The machine learning approach frames this as *nearby sampling*, in which the goal is to sample a molecule with a target property within a structural neighborhood of a reference molecule.

**QED experiment:** For a Quantitative Estimate of Drug-likeness (QED) optimization task [23], the goal is to transfer a molecule with QED $\leq 0.8$ to QED $\geq 0.9$ while maintaining the original structure as much as possible. The similarity to the original is measured by *Tanimoto similarity*, whose threshold is set to $\geq 0.4$. A2A-FM was trained on a 500K ZINC22 subset, where $\mathcal{X}$ represented the latent space of molecular representations [26] and $\mathcal{C} = \mathbb{R}^{32}$ was the space of QED embeddings. Following the same protocol as previous methods [26, 19, 38], we evaluated the algorithm by the success rate of discovering a molecule of desired property (QED, similarity) within the prescribed

| Method | Success (%) |
|---|---|
| DESMILES [34] | 76.9 |
| QMO [19] | 92.8 |
| MolMIM [38] | 94.6 |
| COATI-LDM [25] | 95.6 |
| **A2A-FM (Ours)** | **97.5** |

Table 2: Nearby sampling success rate.

| Method | AUC Values |
|---|---|
| PD+CFG (T=500) | 0.027 |
| PD+CFG (T=300) | 0.450 |
| PD+CFG (T=1000) | 0.060 |
| SI (K=10) | 0.583 |
| OT-CFM | 0.819 |
| **A2A-FM (Ours)** | **0.990** |

Table 3: AUC Values of LogP-TPSA task. PD+CFG is the Partial diffusion method with Classifier Free Guidance with $T$ noise steps.

number of *oracle* sample calls made from the initial molecule (Fig. 6). See Table 2 for the comparison against SOTA [26], and see Appendix B.2 for details and generated samples.

The results in Table 2 show that A2A-FM surpasses SOTA on default oracle calls (50000). Fig. 4 also shows that we achieve much higher success rates across all maximum oracle calls. We note that, unlike QED which is computationally inexpensive to validate, other properties like $\Delta\Delta G$ [45, 1, 29, 50] and vertical ionization potential [47] require costly calculations like DFT and FEP, and biological assays. For such cases, making many oracle calls can be infeasible, so achieving high sampling efficiency is a feat of significant scientific interest.

**LogP&TPSA experiment:** To evaluate the sampling efficiency in all-to-all condition transfer task, we conducted the nearby sampling similar to the QED experiment, except that we chose random 1,024 molecules from ZINC22 as the initial molecules, and aimed at changing their two other properties (*LogP* and *TPSA*) to a randomly selected pair of target values embedded in $\mathcal{C} = \mathbb{R}^{32 \times 2}$. For evaluation, we measured Tanimoto similarity $s_{\mathrm{Tani}}$ and normalized condition error $c_{\mathrm{err}}$ for each transferred sample, and plotted $\Pr(c_{\mathrm{err}} \leq a)$ against $\Pr(s_{\mathrm{Tani}} \leq a)$ to visualize how

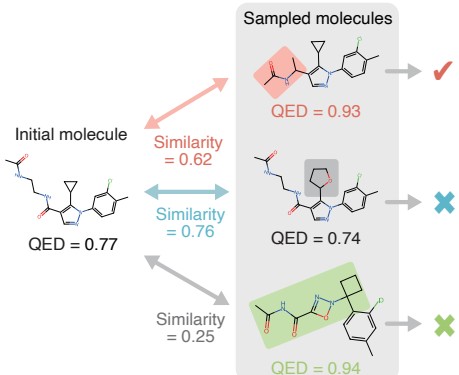

Figure 6: QED optimization task. *An initial molecule is marked success* if there exists a molecule with Tanimoto similarity $\geq 0.4$ and $QED \geq 0.9$ among the molecules sampled by transferring the original. The size of the sample set is called the maximum oracle calls. See Table 2 for the ratio of initial molecules marked success in the dataset of Jin et al. [23].

much we need to trade off $s_{\mathrm{Tani}}$ to increase the probability of sampling molecules within the desired $c_{err}$ threshold. A2A-FM substantially outperformed rivals in AUC (Fig. 5 and Table 3). See also Appendix B.3.

### 5.3 Additional experiments

**Computational cost:** To apply methods like [3, 6] requiring grouped data (Fig. 1 (b), left), data $D = \{(x, c)\}$ is needed to be partitioned into $K$ bins of $M$ samples with similar $c$'s, constrained by $|D| = M \times K$. A choice of large $K$ (small $M$) impairs reliable $P_c$ estimation and, for Multimarginal SI, leads to $O(K^2)$ optimization costs. Meanwhile, small $K$ (large $M$) coarsens the partitions, reducing precision (see Fig. 9, Appendix C). Rising $K$ up to $K = |D|$ is computationally infeasible for Multimarginal SI since it costs $O(K^2)$ for pairwise optimization. In contrast, the computational cost of A2A-FM depends only on $|D|$, which scales identically to OT-CFM (different only in its cost function) and is independent of $K$ for a fixed $|D|$, as shown in Table 6, Appendix C.

**The choice of $\beta$:** As in the case of [9], the selection of the parameter $\beta$ is crucial to the performance. For grouped datasets, $\beta$ can be set sufficiently large (see Section 3.2). However, the same method does not directly apply to the general case due to the trade-off; a large $\beta$ may fail to accurately approximate pairwise OT, while a small $\beta$ can lead to reduced precision in the terminals (Fig. 8, Appendix C). Although the difficulty of this trade-off is a potential limitation of A2A-FM, we empirically observed that the choice of $\beta = N^{1/(2d_c)}$ inspired by [14] was effective in achieving a good balance. We validated this heuristic and assessed the method's robustness to $\beta$ on synthetic non-grouped datasets (Fig. 7, Appendix C). The results confirm that the optimal $\beta$ aligns with our heuristic and demonstrate the method's stability across a range of $\beta$ values spanning *an order of magnitude.* In Appendix A.3, we also dicuss a theretical necessary condition on the rate of $\beta$ that the convergence in 3.1 holds.

## 6 Discussion

**An advantage of using pairwise OTs**: Section 5 demonstrated that A2A-FM performs competitively in transporting $P_{c_1}$ to $P_{c_2}$ in real-world settings. The efficacy of A2A-FM in condition transfer task may be partly explained by its connection to the function representation theorem [30]. This theorem states that if a random variable $X$ has a feature $C = g_c(X)$ with $g_c$ deterministic, then there exists an independent feature $Z = g_z(X)$ such that $(C, Z)$ can generate $X$ (i.e., $X$ decomposes into $C$ and $Z$). When transporting a sample $x_{c_1} \sim P(\cdot | C(X) = c_1) = P_{c_1}(\cdot)$ to $x_{c_2}$, one may leverage the

representation $x_{c_1} = f(z, c_1)$ for an invertible function $f$, and perform transport by mapping $f(z, c_1)$ to $x_{c_2} = f(z, c_2)$ while *keeping the z-component fixed*. This representation is closely related to domain-adversarial training [15], which effectively seeks $z$ in this expression for condition (domain)-invariant inference, and is used in [35] for transporting between conditional distributions. It is also known that, in the case of one-dimensional $x$, the transport of type $f(z, c_1) \mapsto f(z, c_2)$ agrees with the OT with quadratic cost [42, Sec 2.2]. Also, if $\|f(z_1, c_1) - f(z_2, c_2)\| \geq \|f(z_1, c_1) - f(z_1, c_2)\|$ for all $(z_1, c_1)$ and $(z_2, c_2)$, then the OT trivially favors a plan that does not alter $z$. Such a situation may arise when $z$ takes discrete values such that any modification to $z$ results in a larger change in $x$ than modifying $c$. This is also seen in Fig. 3 (a), where each cluster is transported to another cluster.

**Cycle consistency**: It is reasonable to require the consistency property $T_{c_2 \to c_1} \circ T_{c_1 \to c_2} = \text{id}$ for transport maps. In A2A-FM, this can be ensured by enforcing the antisymmetry condition $v_{c_1, c_2} = -v_{c_2, c_1}$ in the model of vector fields. More generally, in some transfer applications, one might prefer the cyclic consistency property: $T_{c_2 \to c_3} \circ T_{c_1 \to c_2} = T_{c_1 \to c_3}$. For instance, some prior works [21, 3] build this cycle consistenct directly into their models. However, A2A-FM may not necessarily enforce this property, because OT does not generally satisfy it. On the other hand, adopting this antisymmetry in QED experiment enhanced the success rate from 94.6% to 97.5%, suggesting some justification for this regularization. While the full effect of this anti-symmetry restriction is a matter of future work, we emphasize that A2A-FM provides a scalable method for non-grouped data, whereas conventional methods with cycle consistency [21, 3] are inapplicable or computationally prohibitive for such data classes (see Appendix 5.3 for details on computational cost).

**A2A-FM on large grouped data:** Many text-labeled image datasets can be regarded as grouped data, as they often provide sufficient data for each independent textual condition. For example, while a sample 'yellow dog' may be rare, there can be many 'dog' samples and 'yellow' samples. As demonstrated in Appendix C, A2A-FM scales effectively to such image datasets. For grouped data, when category-specific classifiers can be trained effectively, classifier-dependent methods are viable options for condition transfer [49, 37]. However, we note that A2A-FM is designed for more general situations in which such classifiers/regressors can be unavailable or unreliable.

## 7  Conclusion

We proposed A2A-FM, an FM-based condition transfer method that can learn pairwise OT from a general data type, including the domain of continuous conditions. To achieve this purpose, we introduced an objective function that realizes pairwise optimal transport in the infinite sample limit. The balance of dataset size and the hyperparameter $\beta$ can pose a limitation, but we provided a stable heuristic whose theory can be a subject of possible future research. We applied A2A-FM to a chemical application of modifying the target attribute of a molecule, demonstrating state-of-the-art performance.

## 8  Acknowledgements

This work was partially conducted during KI's summer internship at Preferred Networks. KF is partially supported by JST CREST JPMJCR2015 and JSPS Grant-in-Aid for Transformative Research Areas (A) 22H05106.

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

# Technical Appendices and Supplementary Material

## A Theories

### A.1 Notations

In this section, we provide the notations that we will be using in the ensuing mathematical formulations and statements.

- $\mathcal{X} \subset \mathbb{R}^{d_x}$: Space of observations.
- $\mathcal{C} \subset \mathbb{R}^{d_c}$: Space of conditions.
- $\mathcal{P}(\mathcal{A})$: the set of all distributions on measure space $\mathcal{A}$.
- $\varpi_{i_1,\ldots,i_k}$: the projection onto $i_1,\ldots,i_k$th component. For example, $\varpi_1 : \mathcal{A} \times \mathcal{B} \to \mathcal{A}$, $\varpi_2 : \mathcal{A} \times \mathcal{B} \to \mathcal{B}$, and $\varpi_{2,3} : \mathcal{A} \times \mathcal{B} \times \mathcal{C} \to \mathcal{B} \times \mathcal{C}$.
- $\Gamma(\mu_1, \mu_2) := \{\Pi \in \mathcal{P}(\mathcal{A} \times \mathcal{A}) \mid \varpi_1 \# \Pi = \mu_1 \text{ and } \varpi_2 \# \Pi = \mu_2\}$ for $\mu_1, \mu_2 \in \mathcal{P}(\mathcal{A})$.
- $Q_{A|B=b} \in \mathcal{P}(\mathcal{A})$: Regular conditional distribution of $Q \in \mathcal{P}(\mathcal{A} \times \mathcal{B})$ where $A, B$ are respectively the random variables on $\mathcal{A}, \mathcal{B}$ with distribution $Q$.
- $P \otimes Q$: For probability distributions $P$ on $\mathcal{X}$ and $Q$ on $\mathcal{Y}$, $P \otimes Q$ denotes the product probability defined by $(P \otimes Q)(A \times B) = P(A)Q(B)$. This gives independent marginals.

### A.2 Convergence to the pairwise optimal transport

Let $\mathcal{P}_2(\mathcal{X})$ denote the space of probabilities with second moments; i.e.,

$$\mathcal{P}_2(\mathcal{X}) := \left\{ P \in \mathcal{P}(\mathcal{X}) \mid \int_{\mathcal{X}} \|x\|^2 dP(x) < \infty \right\}.$$

The 2-Wasserstein distance $W_2(P, Q)$ for $P, Q \in \mathcal{P}_2(\mathcal{X})$ is defined by

$$W_2^2(P, Q) = \inf_{\Pi \in \Gamma(P,Q)} \int_{\mathcal{X} \times \mathcal{X}} \|x - y\|^2 d\Pi(x, y).$$

In the sequel, $\mathcal{P}_2(\mathcal{X})$ is considered to be a metric space with distance $W_2$.

#### A.2.1 Conditional Wasserstein distance

As preliminaries, we first review the conditional Wasserstein distance, which was introduced in [9, 4, 28]. Let $\mathcal{Z}$ be a subset of $\mathbb{R}^r$, which is used as a set of conditions and let $P \in \mathcal{P}(\mathcal{X} \times \mathcal{Z})$. For example, in the setting of conditional generations, $\mathcal{Z} = \mathcal{C}$.

To relate the transport plans for conditional distributions and the plans for joint distributions, we introduce the 4-plans as in [9]. Let $\nu$ be a probability in $\mathcal{P}(\mathcal{Z})$. For $\nu \in \mathcal{P}(\mathcal{Z})$, we define the class of joint distributions with the same marginal on $\mathcal{Z}$:

$$\mathcal{P}(\mathcal{X} \times \mathcal{Z}; \nu) := \{P \in \mathcal{P}(\mathcal{X} \times \mathcal{Z}) \mid (\varpi_2 \# P) = \nu\}.$$

For two probabilities $P, Q \in \mathcal{P}(\mathcal{X} \times \mathcal{Z}; \nu)$, define $\Gamma_\nu^4(P, Q)$ by

$$\Gamma_\nu^4(P, Q) := \{\Pi \in \mathcal{P}((\mathcal{X} \times \mathcal{Z})^2) \mid \varpi_{2,4} \# \Pi = \Delta \# \nu\}, \tag{13}$$

where $\Delta : \mathcal{Z} \to \mathcal{Z} \times \mathcal{Z}$, $z \mapsto (z, z)$, is the diagonal map and $\varpi_{2,4}$ is the projection map to the second and fourth component.

For $P, Q \in \mathcal{P}(\mathcal{X} \times \mathcal{Z}; \nu)$ with finite $p$-th moments, the conditional $p$-Wasserstein distance $W_{p,\nu}(P, Q)$ is defined as follows [9, 28, 4]:

$$W_{p,\nu}(P, Q) := \left( \inf_{\Pi \in \Gamma_\nu^4(P,Q)} \int \|(x_1, z_1) - (x_2, z_2)\|^p d\Pi \right)^{1/p}. \tag{14}$$

It is known [Prop. 1, 9] that the conditional $p$-Wasserstein distance is in fact the average $p$-Wasserstein distances of conditional distributions;

**Proposition A.1.** *Let $(X_1, Z)$ and $(X_2, Z)$ be random variables on $\mathcal{X} \times \mathcal{Z}$ with distributions $P^1$ and $P^2 \in \mathcal{P}(\mathcal{X} \times \mathcal{Z}; \nu)$, which both have finite $p$-th moments. Then,*

$$W_{p,\nu}(P^1, P^2)^p = \int_{\mathcal{Z}} W_p(P_{X_1|Z=z}^1, P_{X_2|Z=z}^2)^p d\nu(z). \tag{15}$$

### A.2.2 Convergence of A2A-FM

Let $\mathcal{D}_N^1 = (x_1^{(i)}, c_1^{(i)})_{i=1}^N$ and $\mathcal{D}_N^2 = (x_2^{(i)}, c_2^{(i)})_{i=1}^N$ be two independent copies of i.i.d. samples with distribution $P \in \mathcal{P}_2(\mathcal{X} \times \mathcal{C})$. Recall that the proposed A2A-FM uses the optimal plan or permutation $\pi_{N,\beta}^*$ which achieves the minimum:

$$\min_{\pi \in \mathfrak{S}_N} \sum_{i=1}^N \|x_1^{(i)} - x_2^{(\pi(i))}\|^2 + \beta\{\|c_1^{(i)} - c_1^{(\pi(i))}\|^2 + \|c_2^{(i)} - c_2^{(\pi(i))}\|^2\}. \tag{16}$$

We formulate $\pi_{N,\beta}^*$ as an optimal transport plan that gives an empirical estimator of the conditional Wasserstein distance. For this purpose, we introduce augmented data

$$\tilde{\mathcal{D}}_1 := \{(x_1^{(i)}, c_1^{(i)}, c_2^{(i)})\}_{i=1}^N \qquad \text{and} \qquad \tilde{\mathcal{D}}_2 := \{(x_2^{(i)}, c_1^{(i)}, c_2^{(i)})\}_{i=1}^N. \tag{17}$$

For notational simplicity, we use $z^{(i)} = (c_1^{(i)}, c_2^{(i)})$, and thus, $\tilde{\mathcal{D}}_1 := \{(x_1^{(i)}, z^{(i)})\}_{i=1}^N$ and $\tilde{\mathcal{D}}_2 := \{(x_2^{(i)}, z^{(i)})\}_{i=1}^N$.

First, it is easy to see that the objective function (16) of the coupling is equal to

$$\min_{\pi \in \mathfrak{S}_N} \sum_{i=1}^N d_\beta\big((x_1^{(i)}, z^{(i)}), (x_2^{(\pi(i))}, z^{(\pi(i))})\big),$$

where the cost function $d_\beta$ on $\mathcal{X} \times (\mathcal{C} \times \mathcal{C})$ is given by

$$d_\beta((x_1, z_1), (x_2, z_2)) = \|x_1 - x_2\|^2 + \beta\|z_1 - z_2\|^2. \tag{18}$$

This means that the minimizer of (16) gives the optimal transport plan between the augmented datasets $\tilde{\mathcal{D}}_1$ and $\tilde{\mathcal{D}}_2$ with the cost $d_\beta$. We express this optimal plan by $\Pi_N^\beta$, an atomic distribution in $\mathcal{P}_2((\mathcal{X} \times \mathcal{C} \times \mathcal{C}) \times (\mathcal{X} \times \mathcal{C} \times \mathcal{C}))$.

Second, consider the following random variables and distributions on $\mathcal{X} \times \mathcal{C} \times \mathcal{C}$;

$$\begin{aligned}(X_1, C_1, C_2) &\sim Q^1 := \varpi_{1,2,4}\#(P \otimes P), \\ (X_2, C_1, C_2) &\sim Q^2 := \varpi_{3,2,4}\#(P \otimes P).\end{aligned} \tag{19}$$

For $Q^1$, the variables $X_1$ and $C_1$ are coupled with the joint distribution $P$, while $C_2$ is independent; for $Q_2$, on the other hand, $X_2$ and $C_2$ are coupled with distribution $P$, and $C_1$ is independent. The datasets $\tilde{\mathcal{D}}_1$ and $\tilde{\mathcal{D}}_2$ are obviously i.i.d. samples with distributiion $Q^1$ and $Q^2$, respectively. Note that the distribution of $Z := (C_1, C_2)$ is equal to $\nu := (\varpi_2\#P) \otimes (\varpi_2\#P) \in \mathcal{P}(\mathcal{C} \times \mathcal{C})$, and common to $Q^1$ and $Q^2$. An important fact is that the conditional distributions given $Z$ satisfy

$$\begin{aligned}Q_{X_1|Z=(c_1,c_2)}^1 &= Q_{X_1|C_1=c_1}^1 = P_{X_1|C_1=c_1}, \\ Q_{X_2|Z=(c_1,c_2)}^2 &= Q_{X_2|C_2=c_2}^2 = P_{X_2|C_2=c_2},\end{aligned} \tag{20}$$

where we use the independence between $(X_1, C_1)$ and $C_2$ for $Q^1$, and a similar relation for $Q^2$.

Let $\widehat{Q}_N^a$ ($a = 1, 2$) be the empirical esitribution of $\tilde{\mathcal{D}}_a$, and $\hat{\nu}_N$ be that of $z^{(i)} = (c_1^{(i)}, c_2^{(i)})$; that is,

$$\widehat{Q}_N^a = \frac{1}{N} \sum_{i=1}^N \delta_{(x_a^{(i)}, Z^{(i)})} \quad (a = 1, 2) \quad \text{and} \quad \hat{\nu}_N = \frac{1}{N} \sum_{i=1}^N \delta_{z^{(i)}} \in \mathcal{P}_2(\mathcal{C} \times \mathcal{C}).$$

Note that, since $z$-component in $\tilde{\mathcal{D}}_1$ and $\tilde{\mathcal{D}}_2$ are identical, the coupling $\Pi_N^\beta$ is a plan in $\Gamma(\widehat{Q}_N^1, \widehat{Q}_N^2; \hat{\nu}_N)$.

With the above preparation, it is not difficult to derive the next result, which shows that, as regularization $\beta$ and sample size $N$ go to infinity, the above empirical optimal plan $\Pi_N^\beta$ converges, up to subsequences, to an optimal plan that gives optimal transport plans for all pairs of conditional probabilities $P_{X|C=c_1}$ and $P_{X|C=c_2}$.

**Theorem A.2.** *Suppose that $\mathcal{X}$ and $\mathcal{C}$ are compact subsets of $\mathbb{R}^{d_x}$ and $\mathbb{R}^{d_c}$, respectively, and that $\beta_k \to \infty$ is an increasing sequence of positive numbers. Let $\Pi_N^{\beta_k}$ be the optimal transport plan in $\Gamma_{\hat{\nu}}^4(\widehat{Q}_N^1, \widehat{Q}_N^2)$ for $W_{2,\beta_k}(\widehat{Q}_N^1, \widehat{Q}_N^2)$, defined as above. Then, the following results hold.*

*(i) There is a subsequence $(N_k)$ so that $\Pi_{N_k}^{\beta_k}$ converges in $\mathcal{P}_2((\mathcal{X} \times \mathcal{Z})^2)$ to $\Pi \in \Gamma_\nu^4(Q^1, Q^2)$ that is an optimal plan for $W_{2,\nu}(Q^1, Q^2)$, where $\mathcal{Z} = \mathcal{C} \times \mathcal{C}$.*

*(ii) If the optimal plan $\Pi$ is identified as a 3-plan $\Gamma_\nu^3(Q^1, Q^2)$, it satisfies*

$$\int_{\mathcal{C}\times\mathcal{C}} \int_{\mathcal{X}\times\mathcal{X}} \|x_1 - x_2\|^2 d\Pi_{c_1,c_2}(x_1, x_2) d\varpi_2 \# P(c_1) d\varpi_2 \# P(c_2)$$

$$= \int_{\mathcal{C}\times\mathcal{C}} W_2^2(P_{X|C=c_1}, P_{X|C=c_2}) d\varpi_2 \# P(c_1) d\varpi_2 \# P(c_2) \quad (21)$$

*where $\Pi_{c_1,c_2} = \Pi_z$ is the disintegration of the 3-plan $\Pi$.*

*(iii) Consequently, $\Pi_{c_1,c_2}$ is the optimal plan to give $W_2(P_{X|C=c_1}, P_{X|C=c_2})$ for $(\varpi_2 \# P) \otimes (\varpi_2 \# P)$-almost every $c_1, c_2$.*

*Proof.* Let $\mathcal{Z} = \mathcal{C} \times \mathcal{C}$. Since $Q^1$ and $Q^2$ have the second moment, the empirical distributions $\widehat{Q}_N^1$ and $\widehat{Q}_N^2$ converge to $Q^1$ and $Q^2$, respectively, in $W_2(\mathcal{X} \times \mathcal{Z})$. Then, from Prop. 12 of Chemseddine et al. [9], for each $k$, there is a subsequence $(N_k)$ such that $\Pi_{N_k}^{\beta_k}$ converges in $W_2$ to an optimal plan $\Pi \in \Gamma_\nu^4(Q^1, Q^2)$ for $W_{2,\nu}(Q^1, Q^2)$ as $k \to \infty$. This proves (i).

For (ii), it follows from Proposition A.1 that

$$\int_{(\mathcal{X}\times\mathcal{Z})^2} \|(x_1, z_1) - (x_2, z_2)\|^2 d\Pi(x_1, z_1, x_2, z_2) = \int_{\mathcal{Z}} W_2^2(Q_{X_1|Z=z}^1, Q_{X_2|Z=z}^2) d\nu(z). \quad (22)$$

Since $z_1 = z_2$ almost surely for $\Pi$, by writing $\Pi$ as an element in $\Gamma_\nu^3(Q^1, Q^2)$, the left hand side of (22) is

$$\int_{\mathcal{X}\times\mathcal{X}\times\mathcal{Z}} \|x_1 - x_2\|^2 d\Pi(x_1, x_2, z) = \int_{\mathcal{Z}} \int_{\mathcal{X}\times\mathcal{X}} \|x_1 - x_2\|^2 d\Pi_z(x_1, x_2) d\nu(z).$$

From (20) and the fact $\nu = (\varpi_2 \# P) \otimes (\varpi_2 \# P)$, the right hand side of (22) is equal to that of (21). Since it holds generally that

$$\int_{\mathcal{X}\times\mathcal{X}} \|x_1 - x_2\|^2 d\Pi_{c_1,c_2}(x_1, x_2) \geq W_2^2(P_{X|C=c_1}, P_{X|C=c_2}), \quad (23)$$

(21) implies that the equality in (23) must hold for $(\varpi_2 \# P) \otimes (\varpi_2 \# P)$-almost every $c_1, c_2$. $\quad\square$

## A.3 Theoritical support on the choice of $\beta$

In this subsection, we derive a necessary condition on the rate of $\beta$ that the convergence of Theorem A.2 holds. Noting that the dimension on the condition space $\mathcal{C} \times \mathcal{C}$ is $2d_c$, we see that the necessary condition below is $\beta = O(N^{1/d_c})$. The choice $\beta = N^{1/(2d_c)}$ used in our experiments satisfies this condition, although it may not be the maximal of the necessary condition.

In the sequel, we discuss the general condition space $\mathcal{C}$ of dimension $d$. Let $\mathcal{X} \subset \mathbb{R}^m$ and $\mathcal{C} \subset \mathbb{R}^d$, and $C$ be a random vector taking values in $\mathcal{C}$ with distribution $P_C$. Let $(X, C) \sim P$ and $(Y, C) \sim Q$ be random vectors taking values in $\mathcal{X} \times \mathcal{C}$, where $P, Q \in \mathcal{P}_2(\mathcal{X} \times \mathcal{C})$ have the same marginal distribution $P_C$, and $X$ and $Y$ are bounded: $\|X\|, \|Y\| \leq M$ almost surely.

Suppose that we have i.i.d. samples $(x_1, c_1), \dots, (x_n, c_n)$ with distribution $P$ and $(y_1, c_1), \dots, (y_n, c_n)$ with distribution $Q$ such that $x_i$ and $y_i$ are conditionally independent given $c_i$ for each $i$. Consider the following empirical conditional OT problem [20, 8, 28]:

$$\sigma_\beta^{(n)} = \arg\min_{\sigma \in \mathfrak{S}_n} F_n(\sigma), \quad F_n(\sigma) := \frac{1}{n} \sum_{i=1}^n \|x_i - y_{\sigma(i)}\|^2 + \beta \frac{1}{n} \sum_{i=1}^n \|c_i - c_{\sigma(i)}\|^2, \quad (24)$$

where $\beta > 0$ is a constant. The optimal transport plan on $(\mathcal{X} \times \mathcal{C}) \times (\mathcal{X} \times \mathcal{C})$ corresponding to $\sigma_* = \sigma_\beta^{(n)}$ is denoted by $\alpha_\beta^{(n)}$: i.e., $\alpha_\beta^{(n)} := \frac{1}{n} \sum_{i=1}^n \delta_{((x_i,c_i),(y_{\sigma_*(i)},c_{\sigma_*(i)}))}$.

Proposition 12 in [8] shows that, for any increasing sequence $\beta_k \to \infty$ ($k \in \mathbb{N}$), there is a subsequence $(n_k)_{k=1}^\infty$ of $\mathbb{N}$ such that, as $k \to \infty$, $\alpha_{\beta_k}^{(n_k)}$ converges w.r.t $W_2((\mathcal{X} \times \mathcal{C}) \times (\mathcal{X} \times \mathcal{C}))$ to the OT plan $\alpha_o \in \Gamma_C^4(P, Q)$ for the following conditional OT problem:

$$\min_{\alpha \in \Gamma_C^4(P,Q)} \int \|x - y\|^2 d\alpha.$$

Here, $\Gamma_C^4(P, Q)$ is the set of 4-plans, defined by

$$\Gamma_C^4(P, Q) := \{\alpha \in \Gamma(P, Q) \mid \pi_\#^{2,4} \alpha = \Delta_\# P_C\},$$

where $\Delta : \mathcal{C} \to \mathcal{C} \times \mathcal{C}, c \mapsto (c, c)$.

We wish to prove $\beta_k = O(n_k^{2/d})$ as $k \to \infty$. For this purpose, we make the following assumptions about the probabilities.

**Assumption 1:** The probability $P_c$ has a density function $f$ with respect to the Lebesgue measure such that there is a constant $B_U > 0$ that satisfies

$$f(c) \leq B_U < \infty \qquad \text{for a.e. } c \in \mathcal{C}.$$

**Assumption 2:**

$$E_C\left[\int \|x - y\|^2 dP_{X|C}(x) dQ_{Y|C}(y)\right] \neq E_C\left[W_2(P_{X|C}, Q_{Y|C})^2\right].$$

The non-equality of Assumption 2 is not restrictive. In fact, by the definition of $W_2$, it generally holds

$$E_C[W_2^2(P_{X|C}, Q_{X|C})] \leq E_C\left[\int \|x - y\|^2 dP_{X|C}(x) dQ_{Y|C}(y)\right].$$

The assumption requires that, for almost all $c$, the conditionally independent ditribution $P_{X|C} \otimes Q_{Y|C}$ does not give $W_2(P_{X|C}, Q_{Y|C})$. In general, the optimal plan with the cost $\|x - y\|^2$ is attained by independent distributions only if the supports of the distributions are orthogonal. In particular, if one of the distributions has a density function with respect to the Lebesgue measure, this is not possible.

We have the following proposition.

**Proposition A.3.** *Let $\beta_k \to \infty$ be an increasing sequence, and $n_k$ be a subsequence of $\mathbb{N}$ such that the optimal plan $\alpha_{\beta_k}^{(n_k)}$ for (24) converges to the conditional OT plan $\alpha_o$ w.r.t. $W_2(\mathcal{X} \times \mathcal{C}) \times (\mathcal{X} \times \mathcal{C})$. Under Assumptions 1 and 2, we have*

$$\beta_k = O\left(n_k^{2/d}\right) \tag{25}$$

*as $k \to \infty$.*

*Proof.* We will derive a contradiction assuming that (25) does not hold. In that case, there is a subsequence $(k')$ of $\mathbb{N}$ such that $\beta_{k'} n_{k'}^{-2/d} \to \infty$ as $k' \to \infty$. W.l.o.g., we assume that the original sequence $\beta_k$ and $n_k$ satisfy

$$\beta_k n_k^{-2/d} \to \infty \qquad (k \to \infty). \tag{26}$$

Take $\gamma_k := \left(\beta_k n_k^{-2/d}\right)^{-1}$, which converges to 0 as $k \to \infty$, and define the radius $r_k > 0$ by

$$r_k := n_k^{-2/d} \gamma_k^{1/2} = \beta_k^{-1/2} n_k^{-1/d}, \tag{27}$$

for which

$$\beta_k r_k = \beta_k^{1/2} n_k^{-1/d} = \gamma_k^{-1/2} \to \infty \quad (k \to \infty) \tag{28}$$

holds. Also, we have $n_k^2 r_k^d = \gamma_k^{d/2} \to 0$, which implies from Proposition A.4 that

$$\mathbb{P}\left(\min_{1 \leq i \leq n_k} \min_{j \neq i} \|c_i - c_j\| \leq r_k\right) \to 0 \qquad (k \to \infty).$$

This means that the probability of the event
$$E_k := \big\{ \|c_i - c_j\| > r_k \text{ for all } 1 \le i < j \le n_k \big\}$$
tends to 1 for $k \to \infty$. Hereafter, we consider a random evnet on $E_k$.

For simplicity, let us write $\sigma_k^* := \sigma_{\beta_k}^{(n_k)}$, and define
$$J_k := \{1 \le i \le n_k \mid \sigma_k^*(i) \ne i\}$$
and $m_k := |J_k|$. It follows that
$$\beta_k \frac{1}{n_k} \sum_{i=1}^{n_k} \|c_i - c_{\sigma_k^*(i)}\|^2 \ge \beta_k \frac{1}{n_k} \sum_{i \in J_k} \|c_i - c_{\sigma_k^*(i)}\|^2 \ge \beta_k \frac{m_k}{n_k} r_k.$$

On the other hand, since $\sigma_k^*$ minimizes $F_{n_k}$, we have
$$\beta_k \frac{1}{n_k} \sum_{i=1}^{n_k} \|c_i - c_{\sigma_k^*(i)}\|^2 \le F_{n_k}(\sigma_k^*) \le F_{n_k}(\mathrm{id}) = \frac{1}{n_k} \sum_{i=1}^{n_k} \|x_i - y_i\|^2 = O_p(1).$$

From the above two inequalities, we have $\beta_k \frac{m_k}{n_k} r_k = O_p(1)$, and thus it holds from (28) that
$$\frac{m_k}{n_k} = O_p((\beta_k r_k)^{-1})) = o_p(1) \quad (k \to \infty). \tag{29}$$

Next, the first term of $F_{n_k}(\sigma_k^*)$ is
$$\int \|x - y\|^2 d\alpha_{\beta_k}^{(n_k)} = \frac{1}{n_k} \sum_{i \notin J_k} \|x_i - y_{\sigma_k^*(i)}\|^2 + \frac{1}{n_k} \sum_{i \in J_k} \|x_i - y_{\sigma_k^*(i)}\|^2$$
$$= \frac{1}{n_k} \sum_{i=1}^{n_k} \|x_i - y_i\|^2 - \frac{1}{n_k} \sum_{i \in J_k} \|x_i - y_i\|^2 + \frac{1}{n_k} \sum_{i \in J_k} \|x_i - y_{\sigma_k^*(i)}\|^2. \tag{30}$$

The second and third terms in the last line are upper bounded by
$$\frac{1}{n_k} \sum_{i \in J_k} 4M^2,$$

which converges to zero in probability as $m_k/n_k = o_p(1)$. As a result, the first term in the last line converges to $E\|X - Y\|^2$ with $X \perp\!\!\!\perp Y | C$ in probability. Because the left hand side of (30) converges to $\int \|x - y\|^2 d\alpha_o$, it implies
$$E\|X - Y\|^2 = E_c\big[W_2^2(P_{X|C}, Q_{X|C})\big],$$

which contradicts Assumption 2. $\qquad\square$

**Proposition A.4.** *Let $X_1, \ldots, X_n$ be i.i.d. $\mathbb{R}^d$-valued random variables with density $f$ on $S \subset \mathbb{R}^d$. Assume that*
$$f(x) \le B_U < \infty \qquad \text{for all } x \in S.$$
*For $r > 0$, let $v_d := \mathrm{Vol}\big(B(0,1)\big)$ be the volume of the unit ball in $\mathbb{R}^d$. Then*
$$\mathbb{P}\left( \min_{1 \le i \le n} \min_{\substack{1 \le j \le n \\ j \ne i}} \|X_i - X_j\| \le r \right) \le \min\left\{ 1, \binom{n}{2} B_U \, v_d \, r^d \right\}.$$

*Proof.* The proof is standard and we omit it. $\qquad\square$

# B  Experimental Details

All of the model training was done using internal NVIDIA V100 GPU cluster.

| Property | Configuration |
|---|---|
| Data space $\mathcal{X}$ | Latent space of [26] ($\mathbb{R}^{512}$) |
| Condition space $\mathcal{C}$ | QED values embedded in $\mathbb{R}^{32}$ same as [25] |
| Training dataset | 500K subset of ZINC22 |
| Initial points $x_1$ at evaluation | 800 reference molecules provided in [23] |
| Initial conditions $c_1$ at evaluation | Embedded QED of $x_1$ |
| Target conditions $c_2$ at evaluation | linspace$(0.84, 0.95, 10)$ (maximum oracle calls $= 50000$) 
 $[0.9, 0.91, 0.92, 0.93, 0.94]$ (otherwise) |

Table 4: Training and evaluation configuration for the QED experiment

## B.1 Synthetic datasets

In the synthetic experiments shown in Fig. 3 (a), we used a dataset sampled from a Gaussian mixture:

$$\frac{1}{6}(\mathcal{N}(x|[0,-1],0.01^2) + \mathcal{N}(x|[0,-3],0.05^2) + \mathcal{N}(x|[\sqrt{3}/2,1],0.01^2) + \mathcal{N}(x|[3\sqrt{3}/2,3],0.05^2)$$
$$+\mathcal{N}(x|[-\sqrt{3}/2,1],0.01^2) + \mathcal{N}(x|[-3\sqrt{3}/2,3],0.05^2)). \tag{31}$$

In the synthetic experiments shown in Fig. 3 (b), we used the dataset consisting of $10^7$ samples and used a version of MLP with residual connection except that, instead of the layer of form $x \to x + \phi(x)$, we used the layer that outputs the concatenation $x \to [x, \phi(x)]$, incrementally increasing the intermediate dimensions until the final output. We used this architecture with width=128, depth=8 for all of the methods. For A2A-FM, we used $\beta = 10$, and we selected this parameter by searching $\beta = 0.01, 0.1, 1, 10, 100$ (see Appendix C for details). The batch size for all the methods were $1 \times 10^3$. In SI, we discretized the conditional space into 5 equally divided partitions. Also in partial diffusion, we used the classifier free guidance method with weight 0.3, used timesteps of $T = 1000$ and reversed the diffusion process for 300 steps. For non-grouped data, the model was trained on one NVIDIA V100 GPU. In both grouped and non-grouped data experiments, we calculated the metric $\mathbb{E}_{x_1 \sim P_{c_1}, c_1, c_2}[\|T_{c_1 \to c_2}(x_1)) - T_{c_1 \to c_2}^{\mathrm{OT}}(x_1))\|^2]$ empirically using 100 i.i.d. samples from the uniform distributions on the support of $c_1, c_2$. For error analysis, we ran this evaluation 10 times and reported the mean and standard deviations of the multiple runs in Table 1.

## B.2 QED experiment

For the task shown in Section 5.2, we trained A2A-FM on a 500K subsampled ZINC22 dataset, and adopted a grid search algorithm similar to [25] (see the pseudocode in Algorithm 2) to evaluate the methods to be compared. We used four NVIDIA V100 GPUs for training. As mentioned in the main manuscript, this search is done for each *initial molecule* by making multiple attempts(Oracle Calls) of the transfer, and the initial molecule is marked *success* if the method of interest succeeds in discovering the molecule satisfying both the Tanimoto similarity and condition range requirement. The initial molecule is marked *fail* if the method fails to find such a transferred molecule within fixed number of oracle calls. We denote the set of initial molecules by $M_0$ in the pseudocode of 2, and we chose this set to be the same set used in [25]. In the grid search approach, we adopted a boosting strategy, which is to scale the estimated velocity field $v$ with a certain parameter $b$ and calculate the ODE using $v_{\mathrm{boosted}} = b \cdot v$. Using this strategy, we enabled our flow-based model to generate various molecules similar to the diffusion models with large guidance weights. For the grid search configurations $(B, C, N)$ in Algorithm 2, we used $B = \text{linspace}(0.8, 2.5, 20), N = \text{linspace}(10^{-3}, 3, 20), C = \text{linspace}(0.84, 0.95, 10)$ when MAX_ORACLE_CALLS $= 50000$. For MAX_ORACLE_CALLS $\in \{25, 100, 500, 1000\}$, we used $B = [1, 2, 3, 4, 5], N = [0], C = [0.9, 0.91, 0.92, 0.93, 0.94]$. We give samples of generated molecules in Fig. 10. We summarize the setting of the QED experiment in Table 4.

In modeling the vector fields $v$, we used a formulation inspired from Isobe et al. [21] in order to reduce the computational cost given by

$$v(\psi_{c_1,c_2}(t), t|c_1, c_2) := \bar{v}(\psi_{c_1,c_2}(t), c(t)|c_2) - \bar{v}(\psi_{c_1,c_2}(t), c(t)|c_1),$$
$$c(t) := c_2 * t + c_1 * (1 - t), \tag{32}$$

**Algorithm 2** Grid search for QED conditioned molecular transfer

**Input:** • Set of initial molecules $M_0 = \{(x, c_{\text{QED}})\}$

  • Boosting weights $B$
  • Noise intensities $N$
  • Target conditions $C$
  • Trained velocity field $v$
  • Number of maximum oracle calls MAX_ORACLE_CALLS.
    # Function for grid searching
 1: **function** GRID_SEARCH($x_0, c_0$)
 2:   $n \leftarrow 1$
 3:   **while** $n \leq$ MAX_ORACLE_CALLS **do**
 4:     **for** $b \in B$ **do**
 5:       **for** $c_1 \in C$ **do**
 6:         **for** $\varepsilon \in N$ **do**
 7:           sample $z \sim \mathcal{N}(0, 1)$, $\hat{x}_0 \leftarrow x_0 + \varepsilon z$
 8:           $x_1 \leftarrow$ ODESolver($b \cdot v(\cdot | c_0, c_1), \hat{x}_0$)
 9:           $n \leftarrow n + 1$
10:           **if** $x_1$ is not decodable **then**
11:             **continue**
12:           **end if**
13:           **if** Tanimoto_similarity($x_0, x_1$) $\geq 0.4$ **and** QED($x_1$) $\geq 0.9$ **then**
14:             **Return:** SUCCESS
15:           **end if**
16:         **end for**
17:       **end for**
18:     **end for**
19:   **end while**
20:   **Return:** FAIL
21: **end function**
    # Loop for all initial points
22: num_success $\leftarrow 0$
23: **for** $(x_0, c_0) \in M_0$ **do**
24:   **if** GRID_SEARCH($x_0, c_0$) = SUCCESS **then**
25:     num_success $\leftarrow$ num_success $+ 1$
26:   **end if**
27: **end for**
28: **Return:** num_success$/|M_0|$

and used the UNet architecture proposed in Kaufman et al. [25] in which we replaced the convolution with dense layers. The network parameters were the same as the ones used in the QED nearby sampling benchmark of Kaufman et al. [25]. This formulation ensures that no transport will be conducted when $c_2 = c_1$ and avoids learning trivial paths. In the training procedure, we first normalized the condition space with the empirical cumulative density functions so that the empirical condition values would become uniformly distributed, and then embedded them using the TimeEmbedding layer to obtain its 32 dimensional representation; this is the same treatment done in Kaufman et al. [25]. We used the latent representation provided in Kaufman et al. [25] and trained A2A-FM on a 500K subset of ZINC22 with batch size=1024, $\beta = (\text{batch\_size})^{1/2d_c} = (1.2419)^{1/2}$, where $d_c = 32$ is the dimension of the conditional space $\mathcal{C}$. For this experiment, we found the choice of $\beta = (\text{batch\_size})^{1/2d_c}$ to perform well, which is the reciprocal to the speed of the convergence of a pair of empirical distributions [14].

### B.3   Logp&TPSA experiment

In the evaluation of methods on LogP-TPSA benchmark (Section 5.2) we used the sampling efficiency curve plotted between (1) normalized discrepancy of LogP and TPSA between generated molecules and the target condition $c_{\text{err}} \in [0, 1]$ and (2) Tanimoto similarity of Morgan fingerprints between

| Property | Configuration |
|---|---|
| Data space $\mathcal{X}$ | Latent space of [26] ($\mathbb{R}^{512}$) |
| Condition space $\mathcal{C}$ | LogP and TPSA values embedded in $\mathbb{R}^{32 \times 2}$ |
| Training dataset | 3.7M subset of ZINC22 |
| Initial points $x_1$ at evaluation | Random samples from ZINC22 |
| Initial conditions $c_1$ at evaluation | Embedded LogP and TPSA of $x_1$ |
| Target conditions $c_2$ at evaluation | $\mathrm{meshgrid}([0, 1, 2, 3, 4], [10, 45, 80, 115, 150])$ |

Table 5: Training and evaluation configuration for the LogP&TPSA experiment

the initial molecule and the generated molecule $s_{\mathrm{Tani}} \in [0, 1]$. Since the decoder of Kaufman et al. [25] does not always succeed in mapping the latent expression to a valid molecule, we made 10 attempts for each target condition by adding different perturbations to the initial latent vector. Detailed procedure for calculating these two metrics for each initial molecule is in Algorithm 3. In our experiments, we used a 3.7M subset from the ZINC22 dataset and created a validation split of initial molecules containing 1024 molecules. We used four NVIDIA V100 GPUs for training. For $\beta$ in A2A-FM, we used $\beta = (1.25)^{1/2}$. For Multimarginal SI, we created 10 clusters in the $\mathcal{C}$ space using the $K$-means algorithm and treated the class labels as discrete conditions. For OT-CFM in this evaluation, we used the full training dataset as $P_\emptyset$ and trained a flow to conditional distributions using the regular OT-CFM framework for conditional generations. Other configurations, including batch size and model architecture, were common among competitive methods and were the same as the QED benchmark. We illustrate samples of generated molecules in Figures 11 to 15. We summarize the training and evaluation configurations in Table 5.

The sampling efficiency curve in Fig. 5 is described mathematically as,

$$y = G(F^{-1}(x)), \tag{33}$$
$$F(x) := \Pr[s_{\mathrm{Tani}} \leq x], \tag{34}$$
$$G(x) := \Pr[c_{\mathrm{err}} \leq x], \tag{35}$$

where $c_{\mathrm{loss}}, s_{\mathrm{Tani}}$ is the output of Algorithm 3, $\Pr[A \leq x]$ is the ratio of members $a \in A$ that satisfies $a \leq x$, and $c_{\mathrm{err}} := c_{\mathrm{loss}}/c_{\mathrm{max}}$ is the normalized $c_{\mathrm{loss}}$ by the normalization factor $c_{\mathrm{max}} = 7$ so that $c_{err} \in [0, 1]$. This way, the curve describes how much one has to trade off the tanimoto similarity threshold (similar to the size of the neighborhood) in order to sample the molecules with $c_{err}$ under a certain error threshold.

## C  Additional experiments

**The robustness of A2A-FM on different $\beta$:** To further analyze the robustness of A2A-FM against the hyperparameter $\beta$ in non-grouped settings, we trained our model with $\beta = 0.01, 0.1, 1, 5, 10, 20, 50, 100$ and calculated the MSE from the ground-truth pairwise OT using the same procedure as Table 1 (b). The result of this experiment is illustrated in Fig. 7, 8. From this experiment, we found out that the hyperparameter $\beta$ is at least robust over a range of one order of magnitude ($\beta \in [1, 10]$). The selection of $\beta$ used in the experiments ($\beta = (\mathrm{batch\_size})^{1/2d_c}$) is within an order of magnitude of this range since we used $\mathrm{batch\_size} = 10^3$, $d_c = 1$ in the non-grouped synthetic data experiment.

**Computational complexity of Multimarginal SI and A2A-FM:** In the synthetic experimental setup depicted in Fig. 3 (b), one might claim that methods that are designed for grouped data, such as Multimarginal SI, could be applied to the setting of general $(x, c)$ datasets by *binning* the dataset. However, this adaptation of methods like Multimarginal SI is challenging in practice. Firstly, as illustrated in Fig. 9, when a small number of bins are used, Multimarginal SI suffers from low transfer accuracy because the partitioning of $\mathcal{C}$ is just simply too coarse. On the other hand, increasing the number of bins not only leads to a substantial increase in computational cost compared to A2A-FM, but also results in a diminished number of data samples per bin, consequently degrading its accuracy of OT estimation (see also Table 6) and Fig. 9.

**CelebA-Dialog HQ 256 Dataset:** To demonstrate the applicability of A2A-FM to high-dimensional grouped data, we trained A2A-FM on the $256 \times 256$ downscaled version of CelebA-Dialog HQ dataset [22] which contains 200K high-quality facial images with 6-level annotations of attributes:

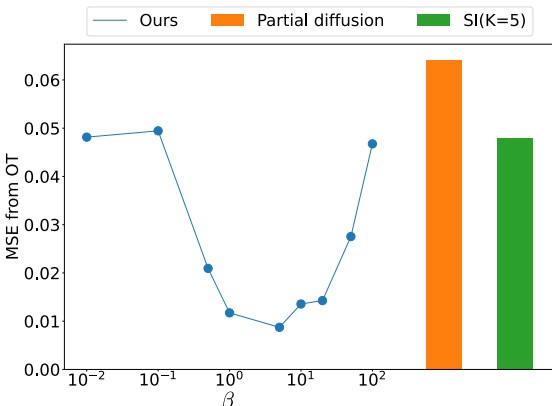

Figure 7: Quantitative results from the additional experiment on the robustness of hyperparameter $\beta$. The results of partial diffusion and multimariginal SI (SI) in the graph are taken from Table 1 (b).

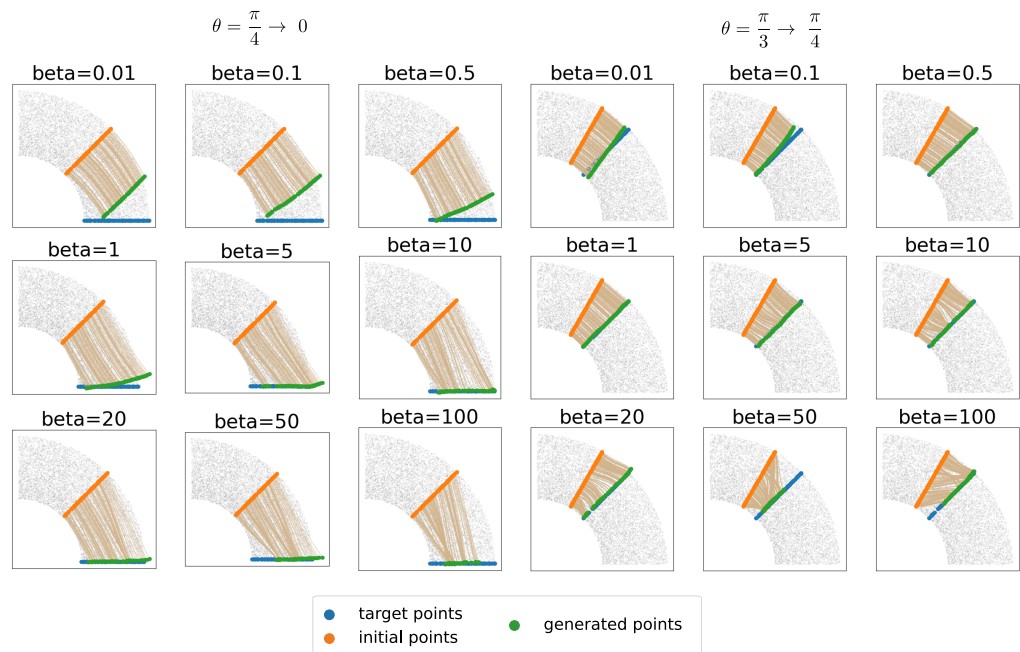

Figure 8: Samples produced in the additional experiment on the robustness of hyperparameter $\beta$.

| Method | MSE from OT | Training time (min) |
|---|---|---|
| SI (K=2) | $1.25 \times 10^{-1}$ | 14.8 |
| SI (K=5) | $4.90 \times 10^{-2}$ | 32.2 |
| SI (K=10) | $2.75 \times 10^{-2}$ | 185.0 |
| parital diffusion | $6.77 \times 10^{-2}$ | 18.1 |
| A2A-FM (Ours) | $1.51 \times 10^{-2}$ | 29.1 |

Table 6: Quantitative results from the additional experiment regarding the change in the number of bins $K$ in Multimarginal SI. Although the performance measured by the MSE from OT improves as $K$ increases, the training time will increase in exchange. Results of partial diffusion and A2A-FM is also reported for comparison. The training of the shown models were done using the same hardware with 1 NVIDIA V100 GPU.

---

**Algorithm 3** Grid search for LogP-TPSA conditioned all-to-all molecular transfer

---

**Input:** • Set of initial molecules $M_0 = \{(x, (c_{\text{LogP}}, c_{\text{TPSA}}))\}$

     • Noise intensity $\varepsilon = 0.1$

     • Target conditions $C = \text{meshgrid}([0, 1, 2, 3, 4], [10, 45, 80, 115, 150])$

     • Trained velocity field $v$

     • Transfer algorithm $\text{Trans}(v, x_0, c_0, c_1)$

     • Condition calculation function $\text{Cond}(\cdot) : \mathcal{X} \to \mathcal{C}$

     # Function for transferring with several attempts

 1: **function** TRANSFER$(x_0, c_0, c_1)$
 2:    c_loss $\leftarrow [0, \ldots, 0]$, similarity $\leftarrow [0, \ldots, 0]$
 3:    **for** $n \in \{1, \ldots, 10\}$ **do**
 4:      sample $z \sim \mathcal{N}(0, 1)$, $\hat{x}_0 \leftarrow x_0 + \varepsilon z$
 5:      $x_1 \leftarrow \text{Trans}(v, x_0, c_0, c_1)$
 6:      **if** $x_1$ is not decodable **or** $\text{Tanimoto\_similarity}(x_0, x_1) = 1$ **then**
 7:        **continue**
 8:      **end if**
 9:      c_loss$[n] \leftarrow \text{MAE}(c_1, \text{Cond}(x_1))$
10:      similarity$[n] \leftarrow \text{Tanimoto\_similarity}(x_0, x_1)$
11:    **end for**
12:    **Return:** $(\min(\text{c\_loss}), \text{similarity}[\text{argmin}(\text{c\_loss})])$
13: **end function**
     # Function to calculate the mean of all-to-all transfer
14: **function** GETMEAN$(x_0, c_0)$
15:    c_loss_mean $\leftarrow 0$, similarity_mean $\leftarrow 0$
16:    **for** $c_1 \in C$ **do**
17:      c_loss$'$, similarity$' \leftarrow$ TRANSFER$(x_0, c_0, c_1)$
18:      c_loss_mean $\leftarrow$ c_loss_mean + c_loss$'$, similarity_mean $\leftarrow$ similarity_mean + similarity$'$
19:    **end for**
20:    **Return:** c_loss_mean$/|C|$, similarity_mean$/|C|$
21: **end function**
     # Loop for all initial points
22: $c_{\text{loss}}$, $s_{\text{Tani}} \leftarrow [0, \ldots, 0]$
23: **for** $i \in \{1, \ldots, |M_0|\}$ **do**
24:    $x_0, c_0 = M_0[i]$
25:    $c_{\text{loss}}[i]$, $s_{\text{Tani}}[i] \leftarrow$ GETMEAN$(x_0, c_0)$
26: **end for**
27: **Return:** $c_{\text{loss}}$, $s_{\text{Tani}}$

---

Bangs, Eyeglasses, Beard, Smiling, Age. In this experiment we used only Beard and Smiling labels and treated these two labels as two-dimensional condition in $[0 : 5]^2$. We used the latent expressions of Rombach et al. [39] and trained a vector field using the UNet architecture with a slight modification for multidimensional conditional inputs. This was done so that we can adopt the independent condition embedding layers to each dimensions of the condition space and to add them to the time embedding as the regular condition embeddings. We display examples of all-to-all transfer results of A2A-FM in Figures 16, 17, 18.

The results of this experiment support our claim that A2A-FM is scalable since the final dimentionality of the latent space was $64 \times 64 \times 3 = 12,288$. We shall note that many text-labeled image datasets are technically grouped databecause computer vision datasets oftentimes contain sufficient data for each independent textual condition, such as 'dog' and 'brown'. In such a case, a classifier for each category may be trained, and condition transfer methods with classifiers like Zhao et al. [49], Preechakul et al. [37] may become a viable option in practice. We stress that A2A-FM is also designed to be able to handle cases in which such good classifiers/regressors cannot be trained. For example, on the aforementioned dataset like ZINC22 with continuous condition, it is difficult to train a high-performance regressor/classifier that can be used for classifier-dependent methods. For

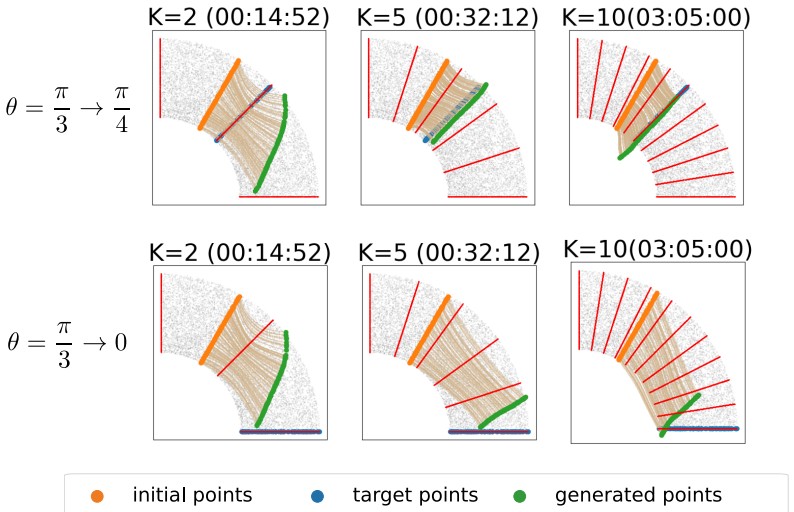

Figure 9: Samples from the additional experiment of multimarginal stochastic interpolants (SI) with different number of bins $K$. The red lines show the boundary of bins. The subtitle of each figure reads as $[K = \#(\text{bins})(\text{trainingtime})]$. This experiment was done using the same hardware with 1 NVIDIA V100 GPU for all models.

example, in Kaufman et al. [26], classifier guidance method for conditional generation does neither perform as well as classifier *free* guidance methods nor its flow-matching counterpart.

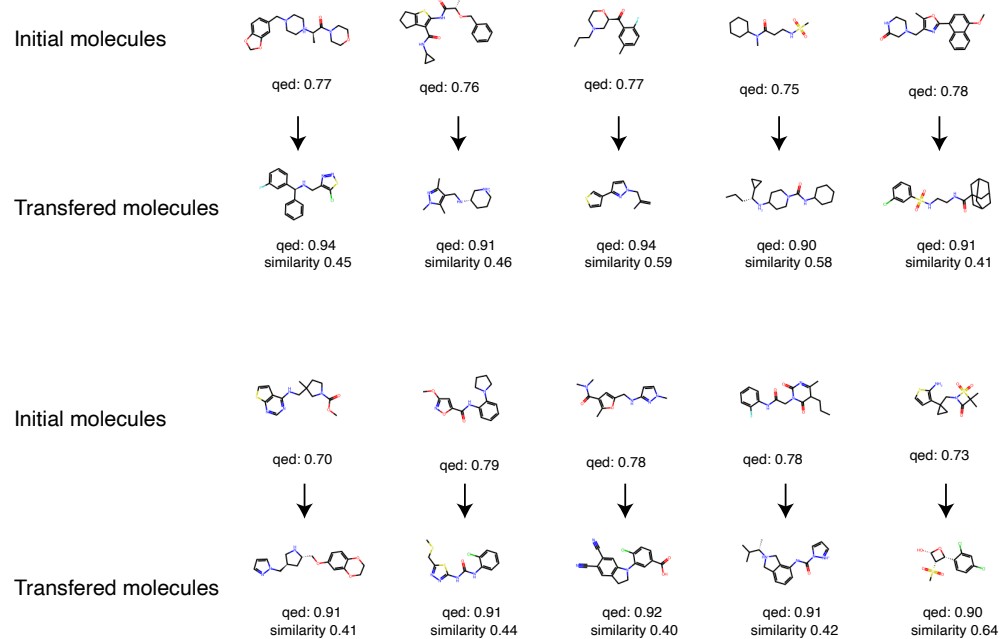

Figure 10: Samples of the nearby sampling in Section 5.2. The first and third row is the initial samples and the second and fourth row are the successful sampled molecules.

Figure 11: All-to-all transfer examples of experiment in Section 5.2.

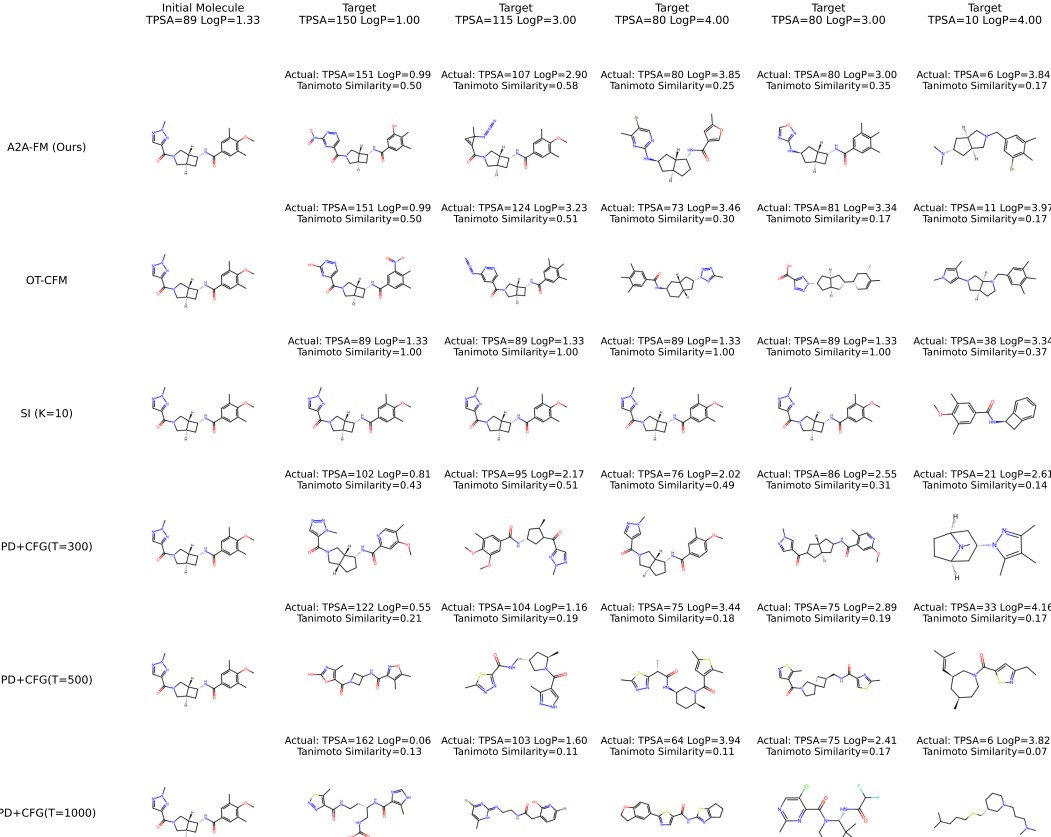

Figure 12: All-to-all transfer examples of experiment in Section 5.2.

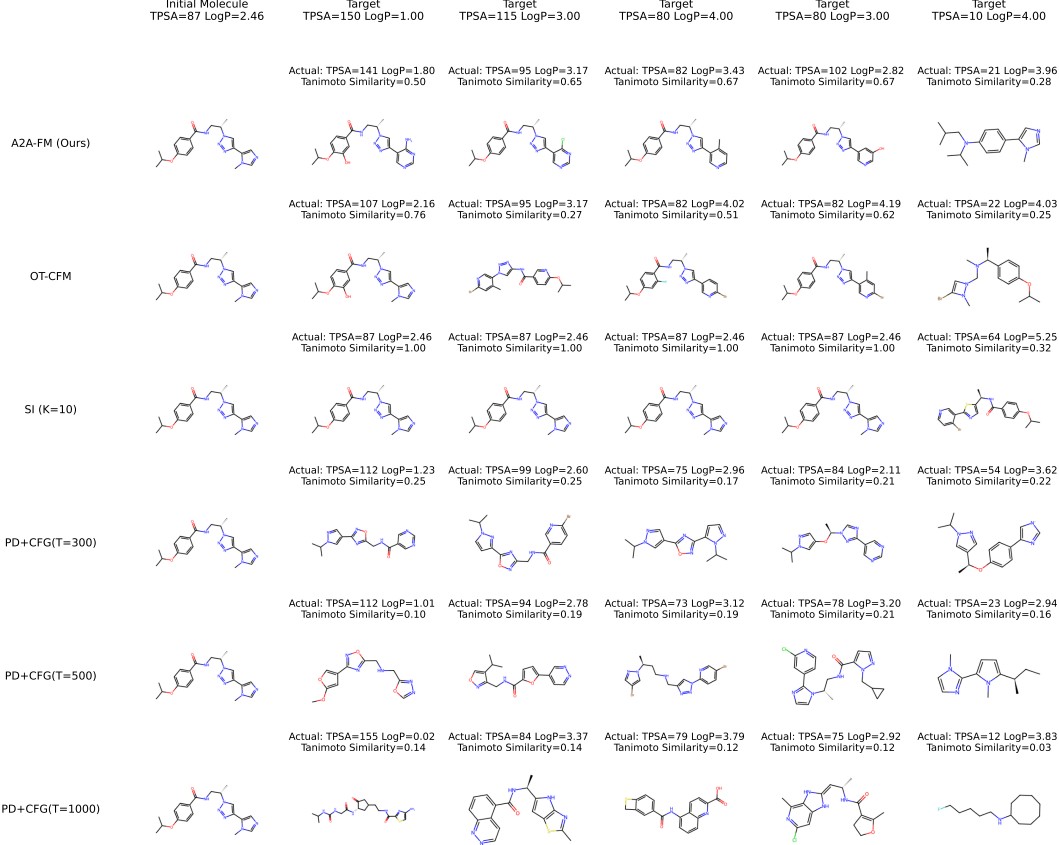

Figure 13: All-to-all transfer examples of experiment in Section 5.2.

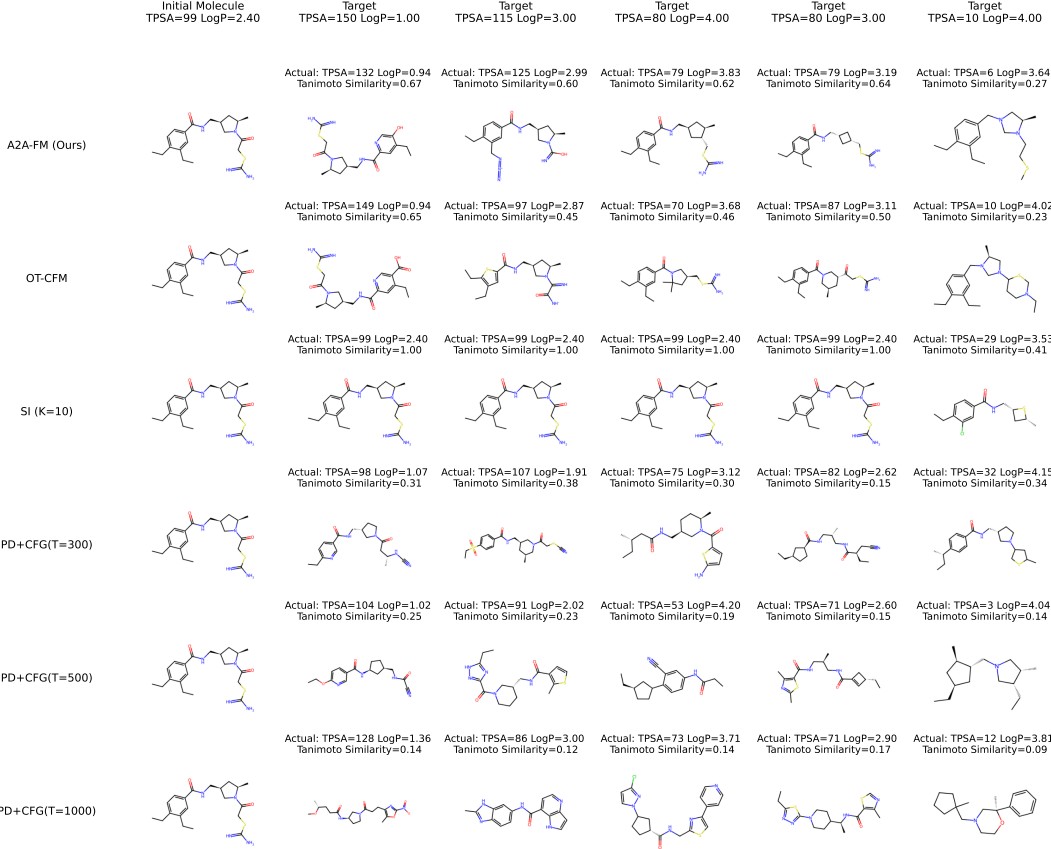

Figure 14: All-to-all transfer examples of experiment in Section 5.2.

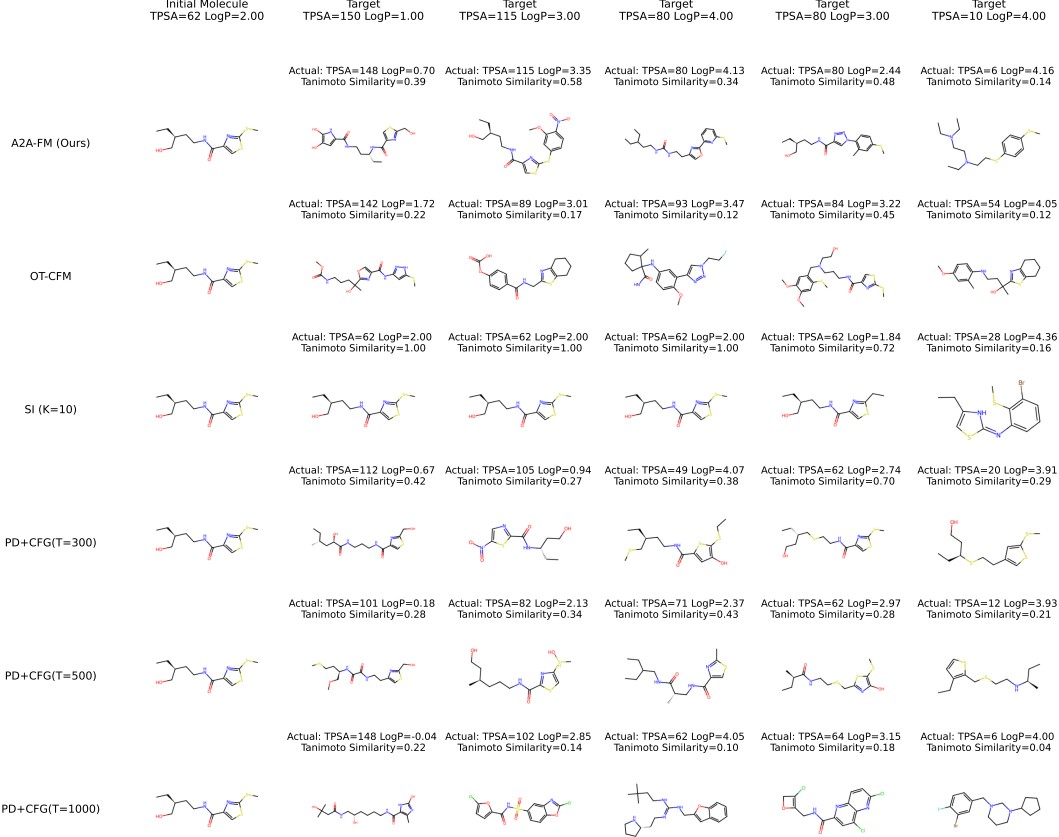

Figure 15: All-to-all transfer examples of experiment in Section 5.2

Beard

original

Smiling

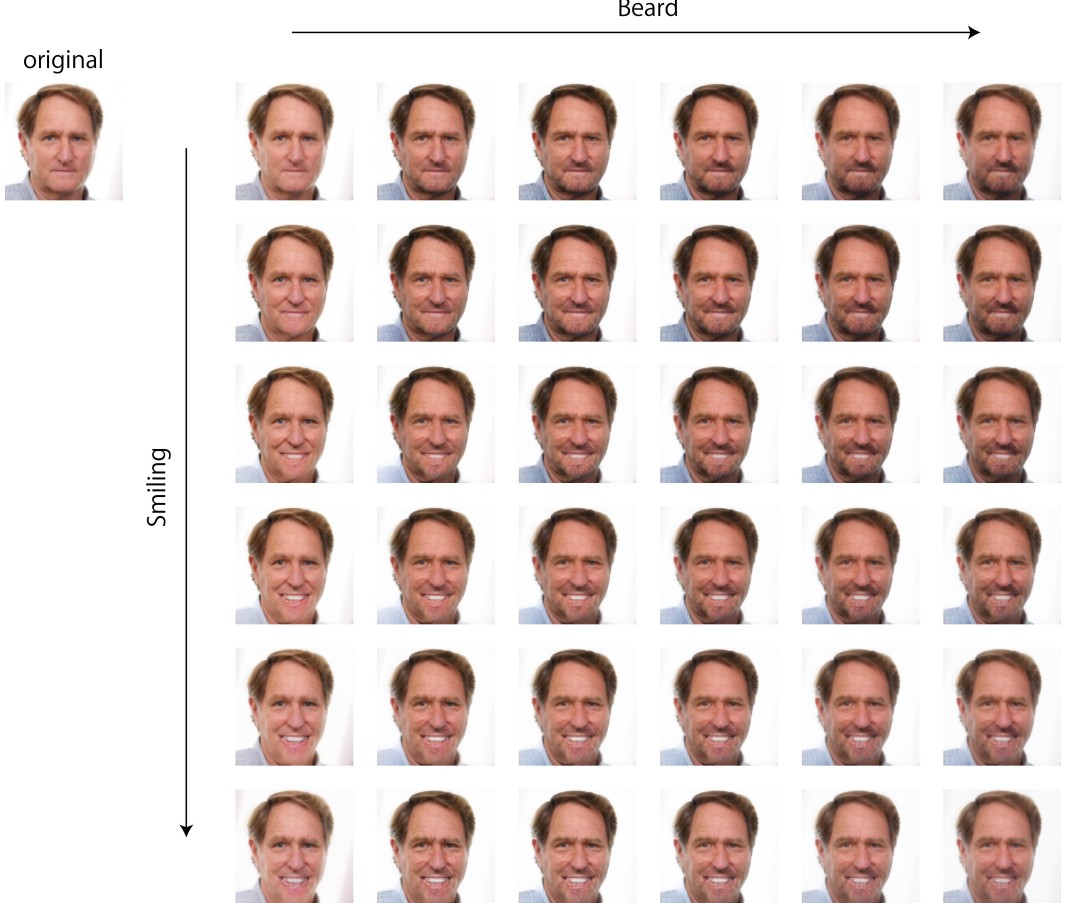

Figure 16: Transfer examples of CelebA-Dialog HQ dataset. A2A-FM is scalable to large scale all-to-all transfer tasks.

Beard

original

Smiling

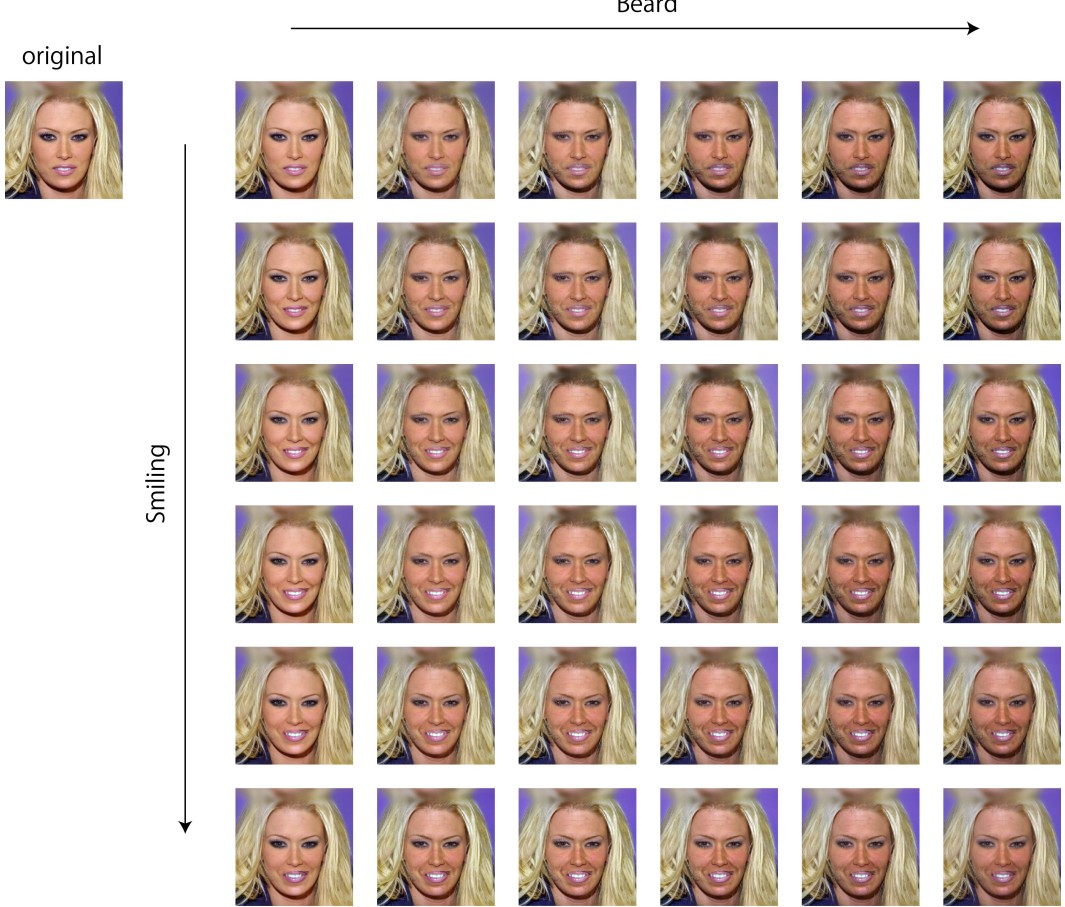

Figure 17: Transfer examples of CelebA-Dialog HQ dataset. A2A-FM is scalable to large scale all-to-all transfer tasks.

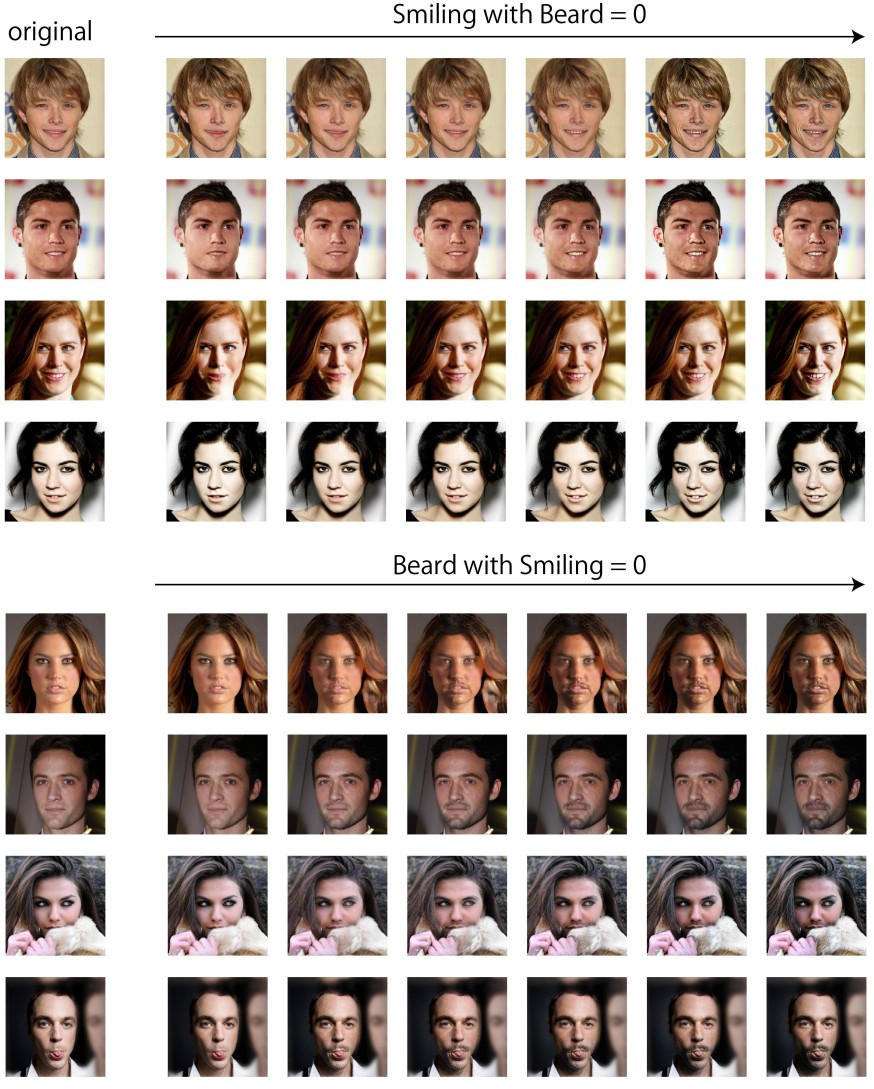

Figure 18: Additional transfer examples of CelebA-Dialog HQ dataset. Here, we fixed the Beard to 0 at changing Smiling attribute and Smiling to 0 at changing the Beard attribute.

