# OpenReview forum: "Pairwise Optimal Transports for Training All-to-All Flow-Based Condition Transfer Model"
_NeurIPS.cc/2025/Conference — NeurIPS 2025 poster_

### Official Review · Reviewer_T9Aq · 2025-06-27

**Clarity:** 1
**Significance:** 3
**Originality:** 3
**Rating:** 4
**Confidence:** 4

**Summary:**

This paper introduces A2A-FM, a flow-based method for learning all-to-all transfer maps between continuous conditional distributions via pairwise optimal transport. The authors propose a novel coupling cost function that integrates data and condition terms, enabling simultaneous learning of OT maps for all condition pairs using minibatch OT-coupled flow matching.

**Questions:**

1.	While the heuristic $\beta$ demonstrates empirical effectiveness, it lacks a theoretical justification. It would be beneficial to understand the sensitivity of performance to $\beta$'s specification, particularly in high-dimensional settings (large d). Are there theoretical bounds that can guide the selection of $\beta$?

2.	In scenarios with highly sparse data (e.g., a large number of unique c values), under what conditions does A2A-FM's approximation of pairwise Optimal Transport (OT) break down? Are there theoretical or empirical guidelines for the minimum number of samples required per condition to ensure reliable approximation?"

3.	Please clarify the application of minibatches to the molecular dataset. What is the strategy for creating minibatches from molecular data?

4.	The discussion of antisymmetry lacks depth. The phrase 'suggesting some justification for this regularization' is unclear. Please provide a more detailed explanation of the rationale behind this regularization and the evidence supporting its effectiveness.

**Ethical Concerns:**

["NO or VERY MINOR ethics concerns only"]

**Final Justification:**

My concerns about the minibatch, antisymmetry, and scenarios on sparse data are addressed. My remaining concern is the clarity of the paper. I would lean to accept a little.

**Limitations:**

As the author said, the balance of dataset size and the hyperparameter $\beta$ can pose a limitation.

**Quality:**

2

**Strengths And Weaknesses:**

_Strengths_:

1.	Tackles the challenge of non-grouped data with continuous conditions (e.g., physical properties in molecules), where existing methods (e.g., Multimarginal SI) fail.

2.	Authors provide convergence analysis proving the approximation of pairwise OT maps..

_Weakness_:

1.	Unclear molecular implementation. Minibatch OT requires fixed-dimensional data, but molecular structures vary in atom count. The paper does not clarify how this dimensional inconsistency was resolved.

2.	While antisymmetry regularization boosts performance, as the author mentioned in Sec. 6, its mechanism and impact lack theoretical or empirical exploration.

3.	In addition to the synthetic data, the manuscript's drug design experiment requires further development. It lacks essential elements, such as a clear introduction to baseline models and a definition of the evaluation metrics used (e.g., AUC). Furthermore, the analysis appears overclaimed, as key properties like $\Delta\Delta G$ and vertical ionization potential were not experimentally tested. The performance of the model on these properties, or others requiring computationally expensive calculations, remains unknown.

---

> ### Author Rebuttal · Authors · 2025-07-25
>
> Thank you very much for the comments and suggestions, we would like to respond to your concerns below.
>
> > Unclear molecular implementation... the paper does not clarify how this dimensional inconsistency was resolved.
>
> We are sorry we were not clear enough.  As we mention in section 5.2, we conducted our algorithm on the latent space used in Kaufmann et al, which provides a Molecular autoencoder based on [23]. The explicit construct of the encoder is given in Section 3.1 and Appendix B of [22], which encodes the SMILES representation of each molecule to 512 dimensional real valued vector.
> The minibatch therefore consists of vectors of dimension 512.
> We used the pretrained encoder provided by [22], which was achieving the SOTA result prior to our research.
>
> > While antisymmetry regularization boosts performance, as the author mentioned in Sec. 6, its mechanism and impact lack theoretical or empirical exploration.
>
> While antisymmetry regularization is not the main focus of this work,  this mechanism is inspired from the solutions of Extended Flow Matching and Stochastic interpolant [2, 18] which both state that , in the general formulation of multimarginal flow, the flow ODE from condition $c_1$ to condition $c_2$ takes the form of
> $$\frac{d}{dt} x_{c_1 \to c_2}(t)  = M(x(t), c(t)) [ \frac{d}{dt}  c(t)  ]    $$
> where $c(0) = c_1,  c(1) = c_2$,  $M$ is a matrix of dimension $d_x \times d_c$.  The theoretical work [18] in particular shows that Dirichlet energy, which is the generalization of the transport cost, is minimized by the flow above with $c(t) = (1-t) c_1 + t c_2$, in which case the flow becomes $M(x(t), c(t))(c_2 - c_1)$.
> While [2,18]  were not able to provide a scalable algorithm in the case of non-grouped data, these two works provide extensive theories behind this flow design.  We will elaborate this point in the revision.
>
> >  ... experiment.. lacks essential elements, such as a clear introduction to baseline models and a definition of the evaluation metrics used (e.g., AUC).
>
> We are sorry that we did not clearly explain them in the main part of the manuscript. To reiterate, our baselines include partial diffusion, stochastic interpolant, and OT-CFM.   We will make sure to include them in the revision.
>
> - Partial diffusion
>
> Partial Diffusion is presented in [22] as well, and it operates the condition transfer from $c$ to $c'$ by  adding a limited noise to the embedded representation of the initial molecule of condition $c$ and apply the classifier guidance with class $c'$ for denoising. We conducted experiments with different noise steps, please see Figure 5.  We used the same architectural and model settings used in [22], please see Appendix C of [22].
>
> - Multi-marginal stochastic interpolant
>
> Multi-marginal stochastic interpolant (SI)[2] is, as described in the related works section, is a framework of constructing a multi-marginal optimal transport based on the interpolation of optimal transport called Generalized Geodesics [5]. As described in section 4,  given the dataset of $P_{c_i}$ and the source distribution $P_\emptyset$, Generalized Geodesics first learns $T_{\emptyset \to c_i}$ empirically with grouped dataset, and generates $P_c$ by first writing $c = \sum \alpha_i c_i$  and using the transport $\sum \alpha_i T_{\emptyset \to c_i}$. SI  approximates this with the flow and  further optimizes the interpolation path by optimizing the path in the $\mathcal{C}$ space by incrementally transforming $c$ with this interpolation. We used the same architectural design as our method for the flow.
>
> - OT-CFM
>
> This is a classic method of [38]. As described in section, we conduct conditional transfer with this method by treating the unconditional distribution as the source distribution.
>
>
> - AUC of figure 5.
>
> We provide this information in the Appendix B.3.  The curves of Fig 5 are the plots between "(1) normalized discrepancy of LogP and TPSA between generated molecules and the target condition cerr ↔[0,1] and (2) Tanimoto similarity of Morgan fingerprints between
> the initial molecule and the generated molecule sTani ↔[0,1]."  Because it is more generally difficult to alter just the desired property while maintaining the similarity to the original molecule, (1) and (2) are in tradeoff relation. To compare the wellness of this tradoff relation, we used "Area Under the Curve (AUC) of the plot of (1) against (2).  We will include this in the main manuscript as well.
>
>
> >  the analysis appears overclaimed, as key properties like $\Delta\Delta G$ and vertical ionization potential were not experimentally tested.
>
> We appreciate reviewer's  mention of these properties. At the same time, *validating* these quantities requires expensive quantum chemistry simulations such as DFT or FEP, whose computational cost on the benchmark dataset of considered size can be infeasbile in academic settings.  For this reason, even many prior *pure chemistry* works (e.g., COATI [23], MolMIM [35]) evaluate on low-cost surrogates like QED or LogP, and we followed the same practice to ensure comparability.  As the general machine learning research, we are providing the results comparable to the state of the art *chemical academic research*.
> In the revision, we make sure that our chemistry experiments are not stand-alone *industrial level* results.
>
>
> > heuristic $\beta$ demonstrates empirical effectiveness, it lacks a theoretical justification.
>
> First we note that, theoretically providing optimal values for $\beta$ is a difficult task. For example, although the application and the coupling cost function differs, Kerrigan et al [24] that was accepted last year also uses the similar parameter, and do not provide an effective heuristics nor a theoretical intuition on how to decide the parameter. We believe that we are the first one to provide this type of heuristics based on the convergence speed of Wasserstein convergence rate and this heuristics itself should be regarded as a contribution.
>
> Although creating precise bounds to obtain optimal values for $\beta$ requires discussions on the convergence rates in Prop.3.1, which is far beyond the scope of this paper, it is possible to observe that the provided heuristics $\beta = N^{1/{2d_c}}$ will fulfill necessary conditions for the convergence. In Prop. 3.1, we stated that the cost function:
>
> $$C(N, \beta) = \min_{\pi \in S_N} \Delta_{\pi}(X) + \beta\Delta_{\pi} (C)$$
>
> where,
> $$\Delta_{\pi} (X) = \frac{1}{N} \sum_{i=1}^N \left[\|x_1^{(i)} - x_2^{(\pi(i))}\|^2\right]$$
> $$\Delta_{\pi}(C) =  \frac{1}{N} \sum_{i=1}^N \left[ \|c_1^{(i)} - c_1^{(\pi(i))}\|^2 + \|c_2^{(i)} - c_2^{(\pi(i))}\|^2\right]$$
> $$ S_N: \text{symmetric group of [1:N]}$$
>
> converges to a finite value under a subsequence $\beta_k, N_k$. On the other hand, from an argument similar to [Yukich2006, section 4.3], with high probability there exists a positive value $K$ such that $ KN^{-1/d_c}  \leq \min_{\pi \in S_N, \pi \neq \operatorname{id}} \|c_1^{(i)}-c_1^{(\pi(i))}\|^2 +\|c_2^{(i)}-c_2^{(\pi(i))}\|^2  $ holds, as this order corresponds to the square of probabilistically high of minimum distance between the nearest neighbors when $N$ instances of $(c_1, c_2)$ are sampled from $R^{d_c + d_c}$.
>  Since the minimizer $\pi$ of $C(N,\beta)$ is not identity maps in regular situations of continuously valued $c$s in the presence of  $\Delta_{\pi} (X)$, this bound is likely to hold for the minimizer of $C(N, \beta)$. At the same time,  when we substitute $\beta = N^{1/{2d_c}}$, we obtain
>
> $$ C(N, \beta) \geq \min_{\pi \in S_N} \Delta_{\pi} (X) + K \underbrace{\beta N^{-1/d_c}}_{N^{-1/(2d_c)}}.$$
>
> Therefore, by choosing this order of $\beta$, we can not only prevent the cost from exploding in the limit of $N\to \infty$,
> but also require $\Delta_{\pi}(C)$ part to approach to $0$ without literally realizing $\pi = id$, thereby facilitating $\Delta_{\pi} (X)$ to learn the coupling between $P(\cdot | c_1)$ and $P(\cdot | c_2)$ for each instance of $(c_1, c_2)$.
> Also,if  $\beta \gg N^{1/{d_c}}$, the cost itself may not converge in the limit of $N\to \infty$.
> The investigation of $\beta$ is an interesting future research question, and  can also help advance the application of Kerrigan et al [24] for which the heuristic has not been established.
>
>
> [Yukich2006] Yukich, Joseph E. Probability theory of classical Euclidean optimization problems. Springer, 2006.
> >  In scenarios with highly sparse data (e.g., a large number of unique c values),  under what conditions does A2A-FM's approximation of pairwise Optimal Transport (OT) break down?
>
> We first note that our method is compatible with dataset in which all c values to appear are unique.  However, just as is the case for all generative models, approximation would break down when the joint data is sparse with respect to the distribution.
> That is, when the set observations of $c$ does not approximate $p_c$, then the model may not capture the continuity pf $p(\cdot | c)$ with respect to $c$.
>
> >  What is the strategy for creating minibatches from molecular data?
>
> After encoding the molecules in the SMILES representation into 512 dimensional vectors with the encoder mentioned above, we sampled the minibatches from randomly shuffled samples, using the standard PyTorch implementation.
>
> >  The discussion of antisymmetry lacks depth.
>
> Please see our comment above.

---

> > ### Comment · Reviewer_T9Aq · 2025-08-05
> >
> > Thank you for the authors' detailed responses. I would increase my score to 4; however, the clarity of the writing remains a minor concern and requires improvement.

---

### Official Review · Reviewer_CVUQ · 2025-06-29

**Clarity:** 2
**Significance:** 2
**Originality:** 3
**Rating:** 4
**Confidence:** 3

**Summary:**

The paper proposes a novel cost function which allows for learning optimal transport (OT) maps between multiple pairs of conditional distributions simultaneously. Then the obtained OT maps are used to couple data points in conditional flow matching analogously to OT-CFM paper (Tong et al., 2023). The resulting model (A2A-FM) can be used to learn translation between conditional distributions with continuous conditions. The authors provide theoretical justifications of the model’s performance and test it on toy examples and on the chemical datasets.

**Questions:**

- Could you please elaborate on the principle of generalized geodesics given in lines 169-170 and its connection to the method of multimarginal Stochastic Interpolants (Albergo et al., 2024?
- Why section 5.1 misses comparison with (Kapusnyak et al., 2024)?
- Why is the hyperparameter $\beta$ not robust to values bigger than $10$ in experiment from section 5.1 in non-grouped settings?

**References.**

Tong, Alexander, et al. "Improving and generalizing flow-based generative models with minibatch optimal transport." Transactions on Machine Learning Research, 2023.

K. Kapusniak, P. Potaptchik, T. Reu, L. Zhang, A. Tong, M. M. Bronstein, J. Bose, and F. Di Giovanni. Metric flow matching for smooth interpolations on the data manifold. In The Thirty-eighth Annual Conference on Neural Information Processing Systems, Nov. 2024.

B. Kaufman, E. C. Williams, C. Underkoffler, R. Pederson, N. Mardirossian, I. Watson, and J. Parkhill. Coati: Multimodal contrastive pretraining for representing and traversing chemical space. Journal of Chemical Information and Modeling, 64(4):1145–1157, 2024.

L. Ambrosio, N. Gigli, and G. Savare. Gradient flows: In metric spaces and in the space of probability measures. Lectures in Mathematics. ETH Zürich. Birkhauser Verlag AG, Basel, Switzerland, 2 edition, Dec. 2008.

M. S. Albergo, N. M. Boffi, M. Lindsey, and E. Vanden-Eijnden. Multimarginal generative modeling with stochastic interpolants. In The Twelfth International Conference on Learning Representations, 2024.

**Ethical Concerns:**

["NO or VERY MINOR ethics concerns only"]

**Final Justification:**

The authors have addressed most of my concerns during the rebuttal phase, thus, I changed my score to positive.

I did not assign higher score since I remain slightly concerned by the experiments - I can not judge the experiment on chemical datasets as I am not an expert in this field, while other experiments are either sunthetic or given only to show the scalability of your approach, i.e. do not demostrate its superior performance.  I remain slightly concerned by the experiments - I can not judge the experiment on chemical datasets as I am not an expert in this field, while other experiments are either sunthetic or given only to show the scalability of your approach, i.e. do not demostrate its superior performance.

Overall, I took into account that other reviewers find the chemical experiment meaningful, and assign slightly positive score, i.e., 4.

**Limitations:**

The authors have addressed the limitations of their approach.

**Paper Formatting Concerns:**

No major concerns, see 'Weaknesses' section for suggestions on clarity improvement.

**Quality:**

3

**Strengths And Weaknesses:**

**Strengths.** The proposed type of cost function (which is used to find the pairwise OT mappings for the family of conditional distributions) allows for dealing with datasets with any type of conditions including continuous ones. The method is supported by the theoretical result showing that it provides good approximations of the ground-truth OT mappings between any pair from the chosen family of conditional distributions (in the limit w.r.t. to the hyper-parameter $\beta$ and number of training samples $N_k$). The method provides good results in the real-world chemical experiment.

**Weaknesses.** Some parts of the paper are hard to understand. ‘Preliminary’ section is missing the overview of definitions from optimal transport theory. This seems to be necessary since later parts of the paper exploit the definitions of OT mappings and OT problem in Kantorovich formulation without defining them previously (lines 110-112). Figure 1 uses captions referring to the methods (Multimarginal SI/EFM) which are not yet defined. The principle of generalized geodesic (Ambrosio et al., 2008) and method of partial diffusion (Kaufman et al., 2024) which are included in comparison on synthetic data (section 5.1, Table 1a,b) are not well explained also.

I am also wondering why section 5.1 misses comparison with (Kapusnyak et al., 2024) included in 'Related work' section. Comparison with this method is needed to understand the position of the proposed approach w.r.t. other methods. The computational inefficiency of (Kapusnyak et al., 2024) stated in line 166 seems to be not a crucial issue for toy experiments.

My other concern is related to the provided theoretical result (Proposition 3.1), specifically, its conditions.  According to this result, the proposed approach allows for learning good approximations of all-to-all OT mappings (between any pair of conditional distributions) in the limit w.r.t. to the number of training samples and hyper-parameter $\beta$, i.e., when $\beta$ is sufficiently large. At the same time, according to the provided experimental details in Appendix C, the authors showed that the parameter $\beta$ is robust up to value $10$ in experiments of section 5.1 in non-grouped settings. It is not evident for me why the hyperparameter is not robust to bigger values, i.e., why the method fails to correctly align the distributions in such cases?

**Overall**, I think that the paper provides interesting cost functions allowing for finding the coupling between multiple pairs of conditional distributions at once. However, the flow-based model by its own is not that novel, it represents an instantiation of the well-known OT-CFM model (Tong et al., 2023) with just another way of finding the coupling. From the practical side,  the approach gives good results in toy experiments as well as the one from the chemistry field, but contains visible artifacts in the experiment with faces in Appendix, see Figure 16. Since I am not an expert in chemistry, it raises questions regarding the overall significance of the proposed approach.

*Typos:*
- line 28 - **in** chemistry
- line 45-46 - **there are** no methods
- line 65 - missing point
- before line 145 - double **the**

---

> ### Author Rebuttal · Authors · 2025-07-27
>
> > ‘Preliminary’ section is missing the overview of definitions from optimal transport theory.
>
> We are sorry that we could not include the introduction to optimal transport theory in the main manuscript of the paper due to page limitation.
> In the revision we will explain the Kantorovich formulation and provide references.
>
> > Figure 1 uses captions referring to the methods (Multimarginal SI/EFM) which are not yet defined.
>
> We are sorry for the confusion, we will provide citations in the captions and also provide reference pointers to the paragraphs explaining these methods.
>
> > The principle of generalized geodesic (Ambrosio et al., 2008) and method of partial diffusion (Kaufman et al., 2024) which are included in comparison on synthetic data (section 5.1, Table 1a,b) are not well explained also.
>
> We are sorry for the lack of explanation. We will explain each one of them below
>
> - Partial diffusion
>
> Partial Diffusion is a method that is presented in [22],  and it operates the condition-transfer from $c$ to $c'$ by (1) adding a limited noise to the embedded representation of the initial molecule of condition $c$ then (2) denoise the perturbed representation with the classifier free guidance for class $c'$. We conducted partial diffusion with different noise steps, please see Figure 5 as well as [22]. In the chemistry experiments, we used the same architectural and model settings used in [22], please see Appendix C of [22] for details.
>
> - Multimarginal Stochastic interpolants and Generalized Geodesics
>
>  As described in section 4, Generalized Geodesics [5], which is the base of Multimarginal stochastic interpolants (SI) [2], is a framework of interpolating the optimal transport. Given the dataset of $P_{c_i}$ and the source distribution $P_\emptyset$, it first learns $T_{\emptyset \to c_i}$ empirically with grouped dataset, and generates $P_c$ by first writing $c = \sum \alpha_i c_i$  and using the transport $\sum \alpha_i T_{\emptyset \to c_i}$.  SI [2] further learns the flow for this tranport, optimizes the interpolation path by optimizing the path in the $C$ space by incrementally transforming $c$ with this interpolation. We used the same architectural design as our method to design the vector flow of [2].
>
> > I am also wondering why section 5.1 misses comparison with (Kapusnyak et al., 2024) included in 'Related work' section.
>
> We are sorry for the confusion. Although Kapusnyak et al. (2024) is correctly cited as Flow matching research that incorporates data manifold their work focuses on unconditional smooth interpolations over data manifold, and does not involve conditioning on external variables or modeling conditional distributions. In contrast, our problem setting is specifically aimed at all-to-all transport between conditional distributions—that is, between  for arbitrary pair of conditions. We acknowledge that our inclusion of this work in the Related Work section may have caused confusion and we will revise that section to more clearly distinguish between unconditional and conditional transport settings.
>
> We did not include the method of Kapusnyak et al for comparison in the submission, since the method designed for unconditional interpolation do not naturally generalize to the current setting. However, it may be possible to adopt Kapusnyak et al. (2024) in conditional generation settings by extending the neural network to accept an additional input for conditions. We experimented this method with this strategy on the LogP-TPSA benchmark mentioned in our paper and obtained:
> ```
> AUC value: 0.179
> ```
> which is a low value compared to other similar methods like OT-CFM. We would like to report that this may be partially due to the instability of the geometry training procedure of Kapusnyak et al., (2024). The time derivative calculation in geodesic path learning led to unstable learning when adopted to high dimensional data like the molecular latents. We need more explorations in the future to adapt the method to the chemical generation scenarios.
>
> > However, the flow-based model by its own is not that novel, it represents an instantiation of the well-known OT-CFM model (Tong et al., 2023) with just another way of finding the coupling.
>
> We would like to respectfully emphasize that our method addresses a substantially different and more challenging task than prior OT-CFM-based approaches. Specifically, as noted by reviewer **uaVd**, *“A2A-FM is one of the first methods to handle non-grouped datasets, where each condition might have only one associated sample. This significantly broadens the applicability of conditional generative models to scientific domains with sparse or continuous condition spaces.”* Unlike previous methods, which often rely on grouped data on finite conditions, our model is designed to operate effectively under non-grouped data, making it applicable to a wider range of real-world scientific settings.
>
> Furthermore, while our method builds upon the OT-CFM framework, many recent advances in flow matching also derive from OT-CFM through alternative coupling mechanisms. For example, [24], published last year, is algorithmically equivalent to OT-CFM, differing only in that it employs a parameterized coupling between joint distributions. Similarly, Riemannian Flow Matching [Chen2023], another widely cited work, modifies CFM primarily by replacing linear interpolation with geodesics.  Although these works, including ours, share a common foundation, developing algorithms and theory capable of accurately learning the target vector field. In our case, we propose a principled and novel approach to simultaneously learn pairwise optimal transport, supported by a rigorous theoretical framework.
>
> We respectfully ask the reviewer to reconsider the novelty and significance of our contribution in terms of the broader scope of applicability and the difficulty of the problem our method addresses.
>
> [Chen2023] Chen, Ricky TQ, and Yaron Lipman. "Flow Matching on General Geometries." The Twelfth International Conference on Learning Representations.
>
> >  From the practical side, the approach gives good results in toy experiments as well as the one from the chemistry field, but contains visible artifacts in the experiment with faces in Appendix, see Figure 16. Since I am not an expert in chemistry, it raises questions regarding the overall significance of the proposed approach.
>
> Thank you for this valuable feedback. We understand your concern that the artifacts in the face experiment could cast doubt on the overall significance of our approach, and we would like to clarify our reasoning. The experiment in the Appendix served a specific and distinct purpose: to verify that our method is scalable to high-dimensional data, a common hurdle in many scientific fields. The model's ability to process this data and produce structured—though not visually perfect—outputs confirms its scalability. The artifacts themselves are a direct consequence of not using any vision-specific architectures, which lies outside the scope of our contribution. Our core results are in the chemistry domain, where the method yields compelling outcomes. Therefore, we respectfully argue that the scalability is well support by the experiment.
>
> >  Could you please elaborate on the principle of generalized geodesics given in lines 169-170 and its connection to the method of multimarginal Stochastic Interpolants (Albergo et al., 2024)?
>
> Please see our comment above.
>
> > Why section 5.1 misses comparison with (Kapusnyak et al., 2024)?
>
> Please see our comment above.
>
> > Why is the hyperparameter $\beta$ not robust to values bigger than $10$ in experiment from section 5.1 in non-grouped settings? It is not evident for me why the hyperparameter is not robust to bigger values, i.e., why the method fails to correctly align the distributions in such cases?
>
> Given *a fixed number of samples $N$*,  indefinitely making the $\beta$ value large in (9) would make the model ignore the $x$ terms and force the choice of $\pi$ for which $c_k^{i} = c_k^{\pi(i)}$ for $k=1,2$.  In the case of *unique $x$ sample per each $c$*, this can cause the permutation $\pi$ to exclusively choose the identity map. As a result, the model would learn the couplings with just a single pair of samples and this can lead to larger errors in OT approximations. When the $\beta$ is of appropriate size with respect to $N$, the learning based on equation (9) would keep the balance of minimizing the transport cost between the set of samples with approximately $c_1$ condition and the set of samples with approximately $c_2$ condition.
>
> Please also see our reply to T9Aq where we derived an inequality that shows for $\beta \gg N^{1/d_c}$, Prop. 3.1 will not converge. Therefore, we can also theoretically state that $\beta \ll N^{1/d_c}$ is necessary.

---

> > ### Comment · Reviewer_CVUQ · 2025-08-05
> > **Additional questions**
> >
> > Thank you for your answers, additional experiment with the method from Kapusnyak et al. (2024) and valuable explanations regarding the parameter $\beta$.
> >
> > >we could not include the introduction to optimal transport theory in the main manuscript of the paper due to page limitation. In the revision we will explain the Kantorovich formulation and provide references.
> >
> > Instead of shortening the background on OT, I kindly suggest you to make an additional section in Appendix.
> >
> > >[experiment in the Appendix] The artifacts themselves are a direct consequence of not using any vision-specific architectures, which lies outside the scope of our contribution.
> >
> > I checked the Appendix, you are using the UNet architecture which usually allow achieving good results in vision-related tasks. I am not sure that the architecture is indeed the main cause of the artifacts. Besides, I have additional questions regarding the claimed scalability of your approach.
> >
> > >The experiment in the Appendix served a specific and distinct purpose: to verify that our method is scalable to high-dimensional data, a common hurdle in many scientific fields. The model's ability to process this data and produce structured—though not visually perfect—outputs confirms its scalability.
> >
> > I checked the description of the experiment one more time and become even more puzzled. According to the line 967 of Appendix, you tested your approach not on the Celeba-Dialoq HQ images themself but on the **latent expressions** of Romach et al. However, latent spaces usually have much lower dimensions than the spaces of images and, thus, the application of the approach on this latent space might not support its scalability. What is the dimension of the latent space which you consider?

---

> > > ### Author Response · Authors · 2025-08-07
> > > **Thank you very much for the response!**
> > >
> > > Thank you very much for your continued engagement and for raising thoughtful concerns.
> > > We would first like to clarify the context of the CelebA-Dialog HQ experiment. As correctly noted, we apply our method in the latent space (of Rombach et al.), which is a standard practice in flow-based generative modeling for high-dimensional image data, as it enhances computational efficiency.
> > > Importantly, the latent space we used has a dimensionality of 64 × 64 × 3 = 12,288, which we believe is sufficiently high to demonstrate the scalability of our approach. While this is not the raw pixel space, it still presents substantial challenges in terms of dimensionality and structure, and we view the successful qualitative results in this setting as a meaningful indicator of general applicability.
> > >
> > >
> > > Regarding the UNet architecture, we fully agree that UNet is commonly employed in vision-related tasks. In our previous comment regarding the certain image region that may appear less refined, our intention was to emphasize that we did not conduct task-specific architectural tuning or employ image-specific techniques such as data augmentation, ActNorm, or variational dequantization, etc. This experiment was not optimized for visual quality, but rather designed to test the general applicability of our method in high-dimensional settings. We will revise the Appendix text to clarify this intent.
> > >
> > > Lastly, we would like to reiterate that our primary focus lies in developing a theoretically justified framework for learning all-to-all OT mappings, particularly for non-grouped conditional distributions. Our main experimental emphasis is on synthetic and scientific datasets—particularly in chemistry—where we believe our contributions have the greatest relevance. The image experiment serves as a qualitative illustration of the method’s generalizability to high-dimensional inputs and is not central to our main claims.
> > >
> > > We hope this clarification addresses your concerns and helps reinforce the broader significance of our contributions.

---

> ### Comment · Reviewer_CVUQ · 2025-08-07
> **Final comment**
>
> Thank you for the answers. Please include the clarifications on the dimension of the latent space in your Appendix. I remain slightly concerned by the experiments - I can not judge the experiment on chemical datasets as I am not an expert in this field, while other experiments are either synthetic or given only to show the scalability of your approach, i.e. do not demostrate its superior performance. However, it seems that other reviewers find your chemical experiment meaningful, thus, I increase my score to 4.

---

### Official Review · Reviewer_uaVd · 2025-06-30

**Clarity:** 4
**Significance:** 4
**Originality:** 4
**Rating:** 5
**Confidence:** 3

**Summary:**

The paper proposes A2A-FM, a flow-based generative modeling framework for learning all-to-all condition transfer maps via pairwise optimal transport. Unlike existing methods that assume grouped datasets with many samples per condition, A2A-FM is designed to handle more general settings, including non-grouped data and continuous condition spaces where each condition may appear only once. The key contribution is a novel coupling cost function that jointly considers data and condition similarity, enabling the learned couplings to converge to the true pairwise optimal transports in the large-sample limit. Integrated within the flow matching framework, these couplings supervise vector fields that transport samples across condition pairs. Empirical evaluations on synthetic and molecular datasets demonstrate that A2A-FM achieves more accurate and efficient condition transfer than prior methods, particularly under the challenging non-grouped setting.

**Questions:**

Proposition 3.1 claims convergence to pairwise OT as \( N \to \infty \) and \( \varpi \to \infty \). However, this is proven under the assumption of joint sampling from \( P(x, c) \).
- How does this convergence guarantee hold when the support of conditional distributions \( P_c \) is non-overlapping or degenerate (e.g., single-point support)?
- More specifically, in the continuous setting, how does the method ensure absolute continuity of \( P_{c_1} \) and \( P_{c_2} \), which is a necessary condition for the existence of OT maps?

The coupling cost mixes sample and condition distances, controlled by \( \varpi \), but the paper offers only a heuristic choice \( \varpi = N^{1/(2d_c)} \).
- Can you derive this scaling from a principled bias-variance decomposition of the coupling error?
- Does this choice minimize some bound on generalization error? If not, what does it optimize?

The model approximates transport maps \( T_{c_1 \to c_2} \) via training a vector field \( v(x, t | c_1, c_2) \).
- Do you assume Lipschitz continuity of the transport maps w.r.t. both \( x \) and \( (c_1, c_2) \)? If so, is this assumption realistic in your non-grouped datasets?

**Ethical Concerns:**

["NO or VERY MINOR ethics concerns only"]

**Final Justification:**

I thank the authors for their thoughtful and detailed rebuttal. After carefully considering their responses and discussions with the other reviewers and the area chair, I am increasing my score to 5. My reasoning is as follows:

Resolved Issues:

* The authors provided satisfactory clarifications to all of my initial concerns, including minibatch OT couplings and regularizations, Lipschitz continuity and bias-variance decomposition.

Remaining Concerns:

* No major concerns remain after the rebuttal. Some minor clarifications could still be made in the final version for improved clarity.

Weighting:

Since all of my core questions were convincingly addressed, and the overall contribution is sound and relevant to the community, I have given significant weight to the improved clarity and completeness offered during the rebuttal.

Overall, I believe this paper makes a meaningful contribution and merits acceptance.

**Limitations:**

Yes

**Quality:**

4

**Strengths And Weaknesses:**

# Strengths

1) A2A-FM is one of the first methods to handle non-grouped datasets, where each condition might have only one associated sample. This significantly broadens the applicability of conditional generative models to scientific domains with sparse or continuous condition spaces.
2) The paper proves that the proposed coupling objective converges to pairwise optimal transport maps in the infinite sample limit, providing a solid mathematical foundation for the method.
3) A2A-FM outperforms prior baselines such as OT-CFM, Multimarginal SI, and partial diffusion on both synthetic and real-world tasks, including challenging molecule optimization benchmarks, demonstrating both accuracy and sampling efficiency.

# Weaknesses

1) Although more efficient than fully multimarginal approaches, the minibatch optimal transport coupling still involves solving a linear assignment problem per iteration, which may become a bottleneck for large batches

2) The success of the method heavily depends on the choice of the coupling regularization parameter. While a heuristic is provided, tuning remains nontrivial, especially for real-world datasets with unknown condition geometry.

---

> ### Author Rebuttal · Authors · 2025-07-28
>
> Thank you very much for the constructive comments, we would like to resolve each one of them below:
>
>
> > Although more efficient than fully multimarginal approaches, the minibatch optimal transport coupling still involves solving a linear assignment problem per iteration, which may become a bottleneck for large batches.
>
> We agree that this could be a bottleneck if the algorithm required very large minibatch to operate.  Interestingly, however,  even in our chemical case of non-grouped dataset of size 500K, we were able to achieve competitive results using a batch size of only 1024, for which the coupling computational cost is negligible.  Although the degree of computational burden may vary depending on the sparsity of the joint distribution, our method does not impose a greater computational cost than any other minibatch based FM algorithms.  We also note that it is possible to cache the optimal couplings before training process or use more efficient OT calculation algorithms (e.g. Sinkhorn) and accelerate the training procedure to mitigate potential bottlenecks in  large scale scenarios.
>
>
> > The success of the method heavily depends on the choice of the coupling regularization parameter. While a heuristic is provided, tuning remains nontrivial, especially for real-world datasets with unknown condition geometry.
>
> Thank you for recognizing the heuristics.  It is true that, when the condition manifold is embedded in very high dimensions, it is difficult to obtain the true intrinsic dimension of the condition. However, when the conditions to control are explainable as variables in $R^{d_c}$ and if all elements in  $R^{d_c}$  are valid conditions, this will not be a problem in the real world settings since we can make use of the provided heuristics.
>
> The difficulty may indeed arise if $P_c$ defined on  some Euclidian space $R^{d}$ which has a very low dimensional support, but such a situation also poses serious challenges in terms of out of distribution generalization for conditions that are possibly not observed.
>
> Moreover, provided that our method is robust against $\beta$ up to an order of magnitude (see Fig. 7 in Appendix) and also that the value of $\beta = N^{1/(2d_c)}$ does not largely change against $d_c$, we think that tuning this parameter is not so difficult in such cases with unknown geometries.
>
> > How does this convergence guarantee hold when the support of conditional distributions ( P_c ) is non-overlapping or degenerate (e.g., single-point support)?
> More specifically, in the continuous setting, how does the method ensure absolute continuity of ( P_{c_1} ) and ( P_{c_2} ), which is a necessary condition for the existence of OT maps?
>
> Indeed, as is required in the well known Benamou-Brenier formula[Benamou 2000], the absolute continuity of $P(\cdot | c_k)$ with respect to the common Lebesgue measure is necessary for the existence of the vector field corresponging to the OT maps.
> However, the results regarding our convergence proof are based on the Wasserstein Distance and OT plan in the form of Kantorovich formulation, for which such assumptions are not required.
> Also, we note that we do not directly use the population-level OT maps in our method. Rather, we construct optimal couplings (i.e., OT plans) between conditional distributions just for the construction of Flow Matching objective. Therefore, the absolute continuity with respect to the Lebesgue measure, which is required for defining continuous OT maps or vector fields in the Benamou–Brenier formulation, is not necessary in our case.
>
>
> [Benamou2000] Benamou, Jean-David, and Yann Brenier. "A computational fluid mechanics solution to the Monge-Kantorovich mass transfer problem." Numerische Mathematik 84.3 (2000): 375-393.
>
> > Can you derive this scaling from a principled bias-variance decomposition of the coupling error?
> > Does this choice minimize some bound on generalization error? If not, what does it optimize?
>
> Thank you very much for the suggestion.  While this is not completely within the scope of this work,  it is an interesting direction for future studies.  Using the notation of the Appendix A.2.2,  we assume the bias-variance decomposition you mentioned can be written as
>
> $$\mathbb{E}[(W^2_{2,\beta}(\hat{Q}^1, \hat{Q}^2)-W^2_{2,\nu}(Q^1, Q^2))^2] = (\mathbb{E}[W^2_{2,\beta}(\hat{Q}^1,\hat{Q}^2)]-W^2_{2,\nu}(Q^1, Q^2))^2 + \operatorname{Var}(W^2_{2,\beta}(\hat{Q}^1,\hat{Q}^2))$$
>
> where, $Q^a$ denotes the augmented data distributions (eq.(17)) and $\hat{Q}^a$ denotes the empirical distribution of $Q^a$. $W_{2,\beta}$ denotes the 2-Wasserstein distance with the $\beta$ -weighted cost function (eq. (16)) and $W_{2,\nu}$ stands for conditional 2-Wasserstein distance (eq. (12), (13)). From Theorem A.2, the left hand side converges to $0$ with an increasing sequence of $\beta_k, N_k$. However, obtaining convergence rates of the two terms in the LHS is non trivial and needs further discussion. Possibly, some kind of justification according to the convergence rates of empirical Wasserstein distances [Fournier2023] may be possible. We would be happy if you can provide any ideas in this direction.
>
> However, for theoretical understanding behind this heuristics, we can show that this selection of $\beta$ meets the necessary condition for the convergence in Prop. 3.1. See our reply to T9Aq. Therefore, although it needs further discussion for finding an optimal value of $\beta$, we can say that our bound has some justification based on necessary condition.
>
> [Fournier2023] Fournier, Nicolas. "Convergence of the empirical measure in expected wasserstein distance: non-asymptotic explicit bounds in ℝd." ESAIM: Probability and Statistics 27 (2023): 749-775.
>
>
>
> >  Do you assume Lipschitz continuity of the transport maps w.r.t. both ( x ) and ( (c_1, c_2) )? If so, is this assumption realistic in your non-grouped datasets?
>
> We do not assume Lipshitz continuity of the transport maps with respect to $( x )$ and $( (c_1, c_2) )$ because our theory does not impose explicit assumption on the model with which to approximate the family of pairwise OTs. Indeed, the existence of OT for each pair $(c_1, c_2)$ does not require  such an assumption.  At the same time, Lipschitz continuity is related to the uniqueness of global solution, as mentioned in Prop 8 of Chemseddine et al.

---

> > ### Comment · Reviewer_uaVd · 2025-08-06
> >
> > I thank the authors for their comprehensive rebuttal. All of my questions have been answered satisfactorily, and I am satisfied with the responses. Therefore, I am increasing my score to 5.

---

### Official Review · Reviewer_26Cp · 2025-07-01

**Clarity:** 3
**Significance:** 3
**Originality:** 3
**Rating:** 5
**Confidence:** 3

**Summary:**

This paper seeks to simultaneously solve a collection of optimal transport problems defined between families of conditional distributions. The authors propose A2A-FM, a method based on flow matching, which uses couplings constructed with a particular cost function such that in theory one recovers the OT plans between the corresponding conditionals. The authors show the convergence of the empirical solution to the OT problem with this cost to the true pairwise OT solution. Experiments are conducted on some synthetic datasets and tasks based on molecular optimization.

**Questions:**

- I found the motivation for the proposed cost (Equation 9) very unclear. While the COT cost is somewhat intuitive (Eqn 3) since it enforces that the condition for a given datapoint is close to the condition of the datapoint it is paired to, this intuition is lost to me in Eqn. 9. It seems that this arises from taking the $i$th datapoint in the first batch $(x_1^i, c_1^i)$ and then appending the value of $c_2^i$, i.e., the conditioning variable for the $i$th datapoint in the second batch.
   - Doesn't this mean that the coupling is sensitive to the ordering of the batches?
   - Is it possible to write out the corresponding cost function in terms of $(x_1, c_1, x_2, c_2)$? Note that as written Equation 9 does not have this form as each term depends on the whole batch and so it is not a proper cost function i.e. depending only on two samples.

**Ethical Concerns:**

["NO or VERY MINOR ethics concerns only"]

**Final Justification:**

I think this is paper is pretty solid in terms of its methodological contributions. The proposed technique is novel, interesting, and a non-trivial extension of existing techniques. The experiments are a clear demonstration of the fact that the methodology works as claimed. I still think that a more diverse set of experiments would strengthen the paper, but I don't think it is a necessary condition for publication of this work.

**Limitations:**

yes

**Quality:**

3

**Strengths And Weaknesses:**

## Strengths:
- The paper works on an interesting and important problem that existing generative models often overlook
- The synthetic experiments are a clear demonstration of the method and that A2A-FM can recover pairwise OT plans
- The method seems to obtain strong performance on the molercular optimization experiments (although I am not an expert on this domain)

## Weaknesses:
- I found the motivation for Eqn 9 quite unclear (see questions section below)
- The experimental settings could be more compelling/varied, particularly for the real-world experiments. While the authors do demonstrate some results on images in the appendix, this is only in the group-conditioning case, and doesn't really demonstrate the key appeal of the method (i.e., the ability to handle continuous conditioning variables).


## Minor
- There are some problems with references throughout.
   - L178: Reference to [32] seems incorrect; I don't see where COT is used in this reference
   -  The references in Section 2.2 are not correct. The authors cite [7] as developing the cost in Eq 3 but this is actually due to [CGS 2008] and also appears in the works [4] and [HHT 2023] which appeared before [7].



[CGS08]: From Knothe’s transport to Brenier’s map and a continuation method for optimal transport. Guillaume Carlier, Alfred Galichon, Filippo Santambrogio, 2008.

[HHT 2023] Conditional optimal transport on function spaces. Bamdad Hosseini, Alexander W. Hsu, Amirhossein Taghvaei, 2023.

---

> ### Author Rebuttal · Authors · 2025-07-29
>
> Thank you very much for constructive comments and suggestions, we would like to respond to each one of them below:
>
>
> > I found the motivation for the proposed cost (Equation 9) very unclear. While the COT cost is somewhat intuitive (Eqn 3) since it enforces that the condition for a given datapoint is close to the condition of the datapoint it is paired to, this intuition is lost to me in Eqn. 9. It seems that this arises from taking the $i$th datapoint in the first batch $(x_1^i, c_1^i)$ and then appending the value of $c_2^i$, i.e., the conditioning variable for the $i$th datapoint in the second batch.
>
> Thank you for the insightful question. You are right to point out that the motivation for "appending the value of $c_2^i$ " is not immediately obvious when compared to the standard COT cost.  In addition to our explanation of intuition in 3.2, let us further clarify on this point in comparison with COT:
>
> In standard COT (Eq. 3), the task is Conditional Generation: "Generate an object with property c." The task is fully defined by a single condition c. Therefore, it's natural to enforce that the paired source and target points share this single condition c.
> In our A2A-FM (Eq. 9), the task is All-to-All Transfer: "Transform an object with property $c_1$ into an object with property $c_2$." To define this transformation, a single condition is not enough. We need a pair of conditions $z = (c_{\rm src}, c_{\rm targ})$, which corresponds to "source condition" and "target condition" respectively.
>
> Given that our task is defined by a condition pair $z$, we use dataset where each data point is associated with such a pair. We start with two independent batches, $B_1 =\{(x_1^{(i)},c_1^{(i)})\} $ and $B_2 =\{(x_2^{(i)},c_2^{(i)})\}$.  For each $i$, this will give rise to pairs $(x_1^{(i)},c_1^{(i)})$, and this is to be coupled with $(x_2^{\pi(i)},c_2^{\pi(i)})$ for some permutation $\pi$.  Here, (Eq. 9) enforces that the $\pi$  does not greatly alter $(c_1^{(i)}, c_2^{(i)})$, so that  $x_1^{(i)} \to x_2^{\pi(i)}$ is approximately mapping from $P( \cdot | c_1^{(i)})$ to $P( \cdot | c_2^{(i)})$.  This way, the triplet $(x_1^{(i)},c_1^{(i)},c_2^{(i)})$ may be read as (source variable, source condition, target condition) where the target condition can be "approximate"--- $(x_1^{(i)},c_1^{(i)}) $ will be coupled to a variable whose condition is approximately $c_1^{(i)}$.
>
> > Doesn't this mean that the coupling is sensitive to the ordering of the batches?
>
> Although we used the order of batches to randomly append conditions by assuming the batches to be randomly shuffled in the implementation, the core of this trick is that the randomly created tuple $(c_1^{(i)}, c_2^{(i)})$ is independently sampled. As mentioned in the Appendix (Equation (18)), the convergence guarantee relies on this independency. If for some reason the dataset cannot be shuffled globally, this independence can be maintained by shuffling within each minibatch before creating the condition pairs.
>
> >  Is it possible to write out the corresponding cost function in terms of $(x_1, c_1, x_2, c_2)$? Note that as written Equation 9 does not have this form as each term depends on the whole batch and so it is not a proper cost function i.e. depending only on two samples.
>
> You are correct in your observation. Equation 9 is not a cost function between two individual samples, but rather the global objective function for the batch-wise optimal transport problem. However, this structure is not unique to our method. It is a standard formulation for any method that utilizes minibatch Optimal Transport to find a coupling, including OT-CFM and COT.
>
> Moreover, parallel to Monge formulation of OT problems as well as COT, it is possible to write down the cost function using joint distributions as
>
> $$ \mathbb{E}\left[ \|X_1-X_2\|^2 + \beta( \|C_1 - C'_1\|^2 + \|C_2 - C'_2\|^2)\right] $$
> where,  random variables $(X_1, (C_1, C_2), X_2, (C'_1, C'_2))$ follows a distribution $\Pi_N \in \Gamma(\hat{Q}^1_N, \hat{Q}^2_N, \hat{\nu}_N)$ defined in the section A.2.2 in the Appendix.
>
> > The experimental settings could be more compelling/varied, particularly for the real-world experiments. While the authors do demonstrate some results on images in the appendix, this is only in the group-conditioning case, and doesn't really demonstrate the key appeal of the method (i.e., the ability to handle continuous conditioning variables).
>
> Sorry for not being able to provide more real-world experiments in group-conditioning cases. We concentrated on scientific application scenarios where condition transfers in continuous domains are more important and well discussed. Moreover the purpose of the additional experiment on image generation in the appendix section is to prove the scalability of the method against high dimensional datasets and we believe that this purpose was achieved by this set of experiments.
>
> > L178: Reference to [32] seems incorrect; I don't see where COT is used in this reference
>
> Thank you for kindly pointing out the mistake in citations. Reference to [32] should be:
>  Manupriya, Piyushi, et al. "Consistent optimal transport with empirical conditional measures." International Conference on Artificial Intelligence and Statistics. PMLR, 2024. We will fix this point in the camera-ready version.
>
> > The references in Section 2.2 are not correct. The authors cite [7] as developing the cost in Eq 3 but this is actually due to [CGS 2008] and also appears in the works [4] and [HHT 2023] which appeared before [7].
>
> Thank you for mentioning the proper origins of Eq. 3. We will mention to the proposed articles in the revision.

---

> > ### Comment · Reviewer_26Cp · 2025-08-01
> >
> > Many thanks for this detailed reply. My first three points above have been adequately addressed -- and I think that some exposition along these lines would help strengthen the clarity of the paper.
> >
> > I do still think that additional experimental settings would help this paper be more impactful. The proposed technique seems quite general and showcasing this across different tasks could only serve to strengthen the submission. That being said, I appreciate that the authors have a particular domain of interest in mind.
> >
> > Overall, I am happy to raise my score to (5) in light of these remarks.

---

> > > ### Author Response · Authors · 2025-08-02
> > >
> > > Thank you very much for your thoughtful consideration of our rebuttal and for your willingness to raise your score. We truly appreciate it.
> > >
> > > We are writing to gently follow up, as it appears the updated score may not have been registered in the system yet. We are currently unable to see the new rating on our end.
> > >
> > > We would be very grateful if you could take a moment to verify that the change was successfully submitted. We apologize for any inconvenience this may cause.
> > >
> > > Thank you again for your time and valuable feedback.

---

> > > > ### Comment · Reviewer_26Cp · 2025-08-02
> > > >
> > > > No worries. I've confirmed the score is updated correctly.
> > > > I believe OpenReview hides the scores from authors after reviewers make their final update.
> > > > You should be able to confirm this on your author console (the score for my review should not be visible there anymore).

---

### Note · Authors · 2025-08-15

In this research, we proposed A2A-FM, a flow based method to simultaneously learn a family of optimal transports between all pairs of conditional distributions, even when the conditional variables are continuous and there is as few as one sample observed for each conditional distribution.

As pointed out by Reviewer uaVd, “the paper proves that the proposed coupling objective converges to pairwise optimal transport maps in the infinite sample limit, providing a solid mathematical foundation for the method. ”  Reviewer 26Cp agrees that “synthetic experiments are a clear demonstration of the method and that A2A-FM can recover pairwise OT plans”.

Moreover, as agreed by most reviewers, the method “provides good results in the real-world chemical experiment”(Reviewer CVUQ). As Reviewer T9Aq agrees, our work "tackles the challenge of non-grouped data with continuous conditions (e.g., physical properties in molecules), where existing methods (e.g., Multimarginal SI) fail."  Reviewer uaVd agrees that our method “outperforms prior baselines such as OT-CFM, Multimarginal SI, and partial diffusion on both synthetic and real-world tasks, including challenging molecule optimization benchmarks, demonstrating both accuracy and sampling efficiency”. These are supported by our SOTA performance in the field.

In the reviewing process there were several concerns, including the depth of the preliminary overviews of the flow-matching and comparative methods as well as the clarity of the details of implementations. There were also requests for scientific motivation behind the heuristics used, including the choice of the hyper parameter for which we were originally just providing ablation studies of its robustness. There was also a request to further strengthen the intuition behind our cost function.

We believe we were able to mostly resolve the raised concerns, and were able to obtain the score-increase from all 4 reviewers. We will add to the revision/Appendix the more elaborate explanations than the ones we provided in the rebuttal. With the revision and the addition to the Appendix based on the discussion, we believe we are able to further strengthen the clarity.  We thank all reviewers for suggestions and constructive comments.

We agree with Reviewer 26Cp that we worked “on an interesting and important problem that existing generative models often overlook”, and it is our hope that our work can contribute to the advancement of flow-based modeling.

---

### Decision · Program_Chairs · 2025-09-17

**Decision:**

Accept (poster)

**Comment:**

The paper extends conditional OT works by considering the case where the conditioned variable could be different in the source and target. Also, unlike most works, grouped data is not assumed. The paper proposes a flow based objective and methodology to solve the corresponding problem. Both theoretically and simulations-wise, the correctness of the methodology is proven. Simulations on a molecular opt benchamrk show that the method may potentially scale to real-world applications.

Overall, the reviews are postive and there is an agreement that the problem is new and the proposed methodology is interesting. The paper seems to reduce a gap in literature in the sense that the previous methods either focus on conditioned variable being the same (COT) or assume grouped data. However, with simple changes in transport-plan parametrization to include two conditioned variables in methods like [1*], may also work. Some discussion or comparison with such simple extensions of existing methods may improve the work.

It seems most of the reviewers initially found the motivation for the basic eqn (9) and other details hard to follow. Even in final comments some reviewers have stressed on clarity. I request the authors to please elaborate on such aspects in their next revision.

[1*] https://proceedings.mlr.press/v238/manupriya24a/manupriya24a.pdf